# The SNAP-25 linker supports fusion intermediates by local lipid interactions

**Ahmed Shaaban[1,2], Madhurima Dhara[3], Walentina Frisch[3], Ali Harb[1], Ali H Shaib[2], Ute Becherer[3], Dieter Bruns[3], Ralf Mohrmann[1,4,5]\***

[1]ZHMB, Saarland University, Homburg, Germany; [2]Department of Molecular Neurobiology, Max Planck Institute for Experimental Medicine, Göttingen, Germany; [3]Institute for Physiology, Center of Integrative Physiology and Molecular Medicine, Saarland University, Homburg, Germany; [4]Institute for Physiology, Otto-von-Guericke University, Magdeburg, Germany; [5]Center for Behavioral Brain Science, Otto-von-Guericke University, Magdeburg, Germany

**Abstract** SNAP-25 is an essential component of SNARE complexes driving fast $Ca^{2+}$-dependent exocytosis. Yet, the functional implications of the tandem-like structure of SNAP-25 are unclear. Here, we have investigated the mechanistic role of the acylated "linker" domain that concatenates the two SNARE motifs within SNAP-25. Refuting older concepts of an inert connector, our detailed structure-function analysis in murine chromaffin cells demonstrates that linker motifs play a crucial role in vesicle priming, triggering, and fusion pore expansion. Mechanistically, we identify two synergistic functions of the SNAP-25 linker: First, linker motifs support t-SNARE interactions and accelerate ternary complex assembly. Second, the acylated N-terminal linker segment engages in local lipid interactions that facilitate fusion triggering and pore evolution, putatively establishing a favorable membrane configuration by shielding phospholipid headgroups and affecting curvature. Hence, the linker is a functional part of the fusion complex that promotes secretion by SNARE interactions as well as concerted lipid interplay.
DOI: https://doi.org/10.7554/eLife.41720.001

**\*For correspondence:**
ralf.mohrmann@med.ovgu.de

**Competing interests:** The authors declare that no competing interests exist.

## Introduction

$Ca^{2+}$-triggered exocytosis is essential for the vesicular release of transmitters, peptides, and hormones. On the molecular level, assembly of membrane-bridging SNARE complexes is believed to induce membrane fusion by forcing vesicle and plasma membrane into close apposition (*Jahn and Fasshauer, 2012*; *Südhof and Rothman, 2009*). SNARE complexes are helix bundles formed by four cognate SNARE motifs, $Q_a$, $Q_b$, $Q_c$, and R, which have been categorized based on sequence homology and the identity of characteristic polar residues in the central interaction layer (*Bock et al., 2001*; *Fasshauer et al., 1998b*). While prototypical SNARE complexes that drive intracellular fusion generally employ four single-pass transmembrane proteins with a single SNARE motif, exocytosis relies on ternary complexes out of one R-SNARE, one $Q_a$-SNARE, and a specialized $Q_{bc}$-SNARE with concatenated $Q_b$ and $Q_c$ motifs (*Weimbs et al., 1998*). In neurons and neuroendocrine cells, the set of exocytotic SNAREs typically comprises synaptobrevin (syb)−2 (R), syntaxin (stx)−1A ($Q_a$), and SNAP-25 ($Q_{bc}$).

The two SNARE motifs in $Q_{bc}$ proteins are connected by a 'linker' peptide, whose length varies between different isoforms. Although all $Q_{bc}$ SNAREs lack a transmembrane domain, several isoforms are membrane-anchored by acylated cysteine residues in the N-terminal linker region. In SNAP-25a and b, the principal isoforms in neurons and neuroendocrine cells, a cluster of 4 cysteines forms a palmitoylation site (*Hess et al., 1992*; *Lane and Liu, 1997*; *Veit et al., 1996*). While SNAP-25 is efficiently acylated by members of the DHHC acyltransferase family (*Greaves et al., 2009*;

*Greaves et al., 2010*), not all cysteines seem to be modified in vivo (*Foley et al., 2012*). Interestingly, membrane association is facilitated by stx-1A interactions, which likely promotes lipidation by membrane-bound acyltransferases (*Gonelle-Gispert et al., 2000*; *Vogel et al., 2000*). Yet, a SNAP-25 linker fragment (aa 85–120) lacking both SNARE motifs can still be effectively acylated (*Gonzalo et al., 1999*), likely owing to transient membrane interactions via hydrophobic and positively charged residues (*Greaves et al., 2009*; *Weber et al., 2017*). Mutation of one or more cysteine residues results in a correlated reduction in plasma membrane localization of SNAP-25 (*Gonelle-Gispert et al., 2000*; *Nagy et al., 2008*; *Washbourne et al., 2001*). Acylation-deficient SNAP-25 mutants were inconsistently reported to be dysfunctional (*Washbourne et al., 2001*), to partially drive exocytosis (*Gonelle-Gispert et al., 2000*), or to primarily change release kinetics (*Nagy et al., 2008*).

Site-directed spin-labelling experiments indicated that the SNAP-25 linker is mostly unstructured (*Margittai et al., 2001*) and thus may act as a simple connector. This view was fostered by data that the linker is only loosely associated with the SNARE complex, from which the N-terminal linker segment can be proteolytically cleaved (*Fasshauer et al., 1998a*). Moreover, the linker domain is dispensable for liposome fusion in vitro (*Parlati et al., 1999*) and the physical linkage of SNARE motifs seems inconsequential for secretion in cracked Botulinum toxin (BoTn) /E treated PC12 cells, as infusion of a toxin-resistant $Q_c$ motif can reinstate release (*Chen et al., 1999*). However, rescue efficiency of complementing SNAP-25 fragments in cellular assays seems to strictly depend on experimental conditions, as another study reported a recovery of secretion only in permeabilized cells after initial rundown of release (*Wang et al., 2008*). Contradicting the notion of a dispensable linker, functional differences between the $Q_{bc}$ isoforms SNAP-25 and SNAP-23 have recently been attributed to a short sequence variation within the linker (*Nagy et al., 2008*).

Here, we have studied the role of the SNAP-25 linker domain in a systematic structure-function analysis. Starkly objecting the prevalent notion of an inert connector, our new results establish that the linker peptide is a vital component of SNARE complexes that serves multi-tiered mechanistic functions during fast $Ca^{2+}$-dependent secretion. In particular, we present first evidence that the linker establishes local lipid:SNARE contacts, which may critically support lipidic fusion intermediates in late mechanistic stages of the exocytotic process.

## Results

### The SNAP-25 linker is a functional component of the fusion machinery

Earlier work suggested that the physical linkage of the two SNARE motifs in SNAP-25 is not a strict prerequisite of transmitter release (*Chen et al., 1999*; *Wang et al., 2008*). As former studies exclusively relied on population-based release assays with limited time resolution, we wondered whether expression of unlinked SNAP-25 fragments could fully reestablish the kinetic properties of fast secretion. To investigate potential kinetic deficits, we setup a reconstitution system based on $Snap25^{-/-}$ (KO) chromaffin cells, wherein two complementing SNAP-25 fragments (each containing one SNARE motif) were co-expressed using a bicistronic Semliki Forest (SF) virus. $Ca^{2+}$-dependent secretion in infected KO cells was analyzed by capacitance measurements and amperometry in combination with $Ca^{2+}$-uncaging (*Mohrmann et al., 2013*), allowing for the analysis of secretion kinetics at the millisecond scale. Due to an unforeseen instability of isolated SN2 fragments in chromaffin cells (*Figure 1—figure supplement 1A*), we used a mCherry (mCh)-SN2 fusion protein in all rescue paradigms. That said, the N-terminal mCh-tag is not expected to interfere with the function of SN2, since SNAP-25-based FRET probes carrying a fluorophore in a similar position exhibited normal interactions with cognate SNAREs (*An and Almers, 2004*; *Wang et al., 2008*). Strikingly, co-expression of a fragment covering SN1 and the full linker (SN1-L; aa 1–141) together with mCh-SN2 (aa 138–206) only resulted in a poor reconstitution of the secretory response to a brief increase of $[Ca^{2+}]_i$ (~30 µM) in KO cells (*Figure 1Aa,b*), albeit both fragments were overexpressed at high levels according to our Western Blot analysis (*Figure 1—figure supplement 1B*). Moreover, release mediated by isolated SNAP-25 fragments under these conditions was kinetically delayed compared to controls expressing wildtype (WT) SNAP-25, as seen from a significant increase in the time constant of monoexponential fits (*Figure 1Ab*) of $\Delta C_M$. To better compare our rescue data with the original results of *Chen et al. (1999)*, we also expressed SN25Δ26 (aa 1–180) together with mCh-SN2 in KO

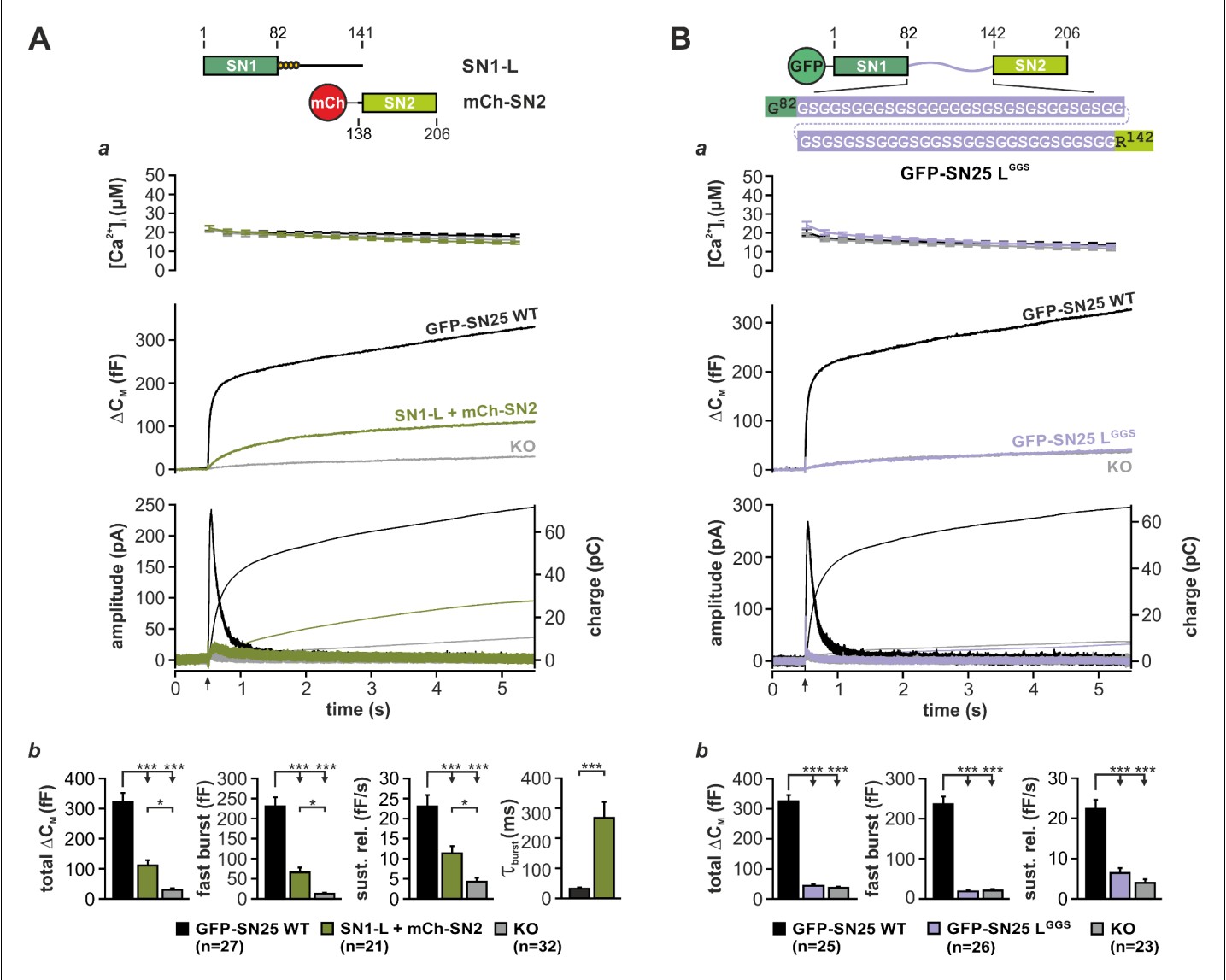

**Figure 1.** Integrity and primary structure of the SNAP-25 linker are important for $Ca^{2+}$-dependent secretion. (**A**) Cartoon (*top*) illustrates the structure of the two co-expressed SNAP-25 fragments: SN1-L (aa 1–141) comprises SN1 and the full linker, while mCherry (mCh)-SN2 represents SN2 (aa 138–206) N-terminally fused to mCherry. (**a**) Electrophysiological measurements of secretion in KO chromaffin cells co-expressing mCh-SN2 and SN1-L showed a poor rescue of secretion. Averaged traces for $[Ca^{2+}]_i$ (*top*), capacitance measurements (*middle*) and amperometric recordings (*bottom*) in KO cells (gray) and in KO cells expressing the fragments (light green) or WT protein (black) are shown. Arrow indicates UV-flash. (**b**) Quantification of the data presented in (**a**). Average capacitance changes after 5 s (*total $\Delta C_M$*) and 1 s (*fast burst*) are depicted. Sustained release represents mean $\Delta C_M$/s occurring between 1 s and 5 s. Kinetics of the secretory burst was approximated by a monoexponential function, yielding $\tau_{burst}$. (**B**) Cartoon (*top*) illustrates the substitution of the whole linker (aa 83–141) by a flexible G/S peptide in SN25 $L^{GGS}$. (**a**) Characterization of GFP-SN25 $L^{GGS}$-mediated secretion in KO chromaffin cells by $Ca^{2+}$-uncaging. Depicted are averaged traces for $[Ca^{2+}]_i$ (*top*), capacitance measurements (*middle*) and amperometric recordings (*bottom*) in KO cells (gray) and KO cells expressing GFP-SN25 $L^{GGS}$ (pastel blue) or WT protein (black). (**b**) Quantitative analysis of the average capacitance changes after 5 s (*total $\Delta C_M$*) and 1 s (*fast burst*). Sustained release represents mean $\Delta C_M$/s occurring between 1 s and 5 s. Data are shown as mean ± SEM. Statistical analysis was done with ANOVA and Tukey's test.

DOI: https://doi.org/10.7554/eLife.41720.002

The following source data and figure supplements are available for figure 1:

**Source data 1.** Extended statistical data as Microsoft Excel spreadsheet.

DOI: https://doi.org/10.7554/eLife.41720.009

**Figure supplement 1.** Reconstitution experiments with two complementary SNAP-25 fragments.

DOI: https://doi.org/10.7554/eLife.41720.003

**Figure supplement 1—source data 1.** Extended statistical data as Microsoft Excel spreadsheet.

*Figure 1 continued on next page*

*Figure 1 continued*

DOI: https://doi.org/10.7554/eLife.41720.004

**Figure supplement 2.** GFP-SN25 L$^{GGS}$ is unable to drive secretion in response to strong, long-lasting stimuli in KO cells but exerts a dominant-negative effect in WT cells.

DOI: https://doi.org/10.7554/eLife.41720.005

**Figure supplement 2—source data 1.** Extended statistical data as Microsoft Excel spreadsheet.

DOI: https://doi.org/10.7554/eLife.41720.006

**Figure supplement 3.** Expression analysis of SNAP-25 linker mutants.

DOI: https://doi.org/10.7554/eLife.41720.007

**Figure supplement 3—source data 1.** Extended statistical data as Microsoft Excel spreadsheet.

DOI: https://doi.org/10.7554/eLife.41720.008

chromaffin cells (*Figure 1—figure supplement 1C*). Expression of SN25Δ26 instead of the shorter SN1-L fragment did however not enhance but clearly suppress secretion (*Figure 1—figure supplement 1Ca,b*), replicating a known dominant-negative action of SN25Δ26 (*Sørensen et al., 2006*). In sum, our data thus point to a yet unacknowledged mechanistic requirement for SNAP-25 linker continuity in secretion.

To evaluate the prevalent view of the linker as a functionally dispensable connector, we next generated a mutant SNAP-25 variant (denoted SN25 L$^{GGS}$), in which the entire linker was substituted by a flexible, equally-sized peptide. Following well-established design principles for inert 'linker' peptides in fusion proteins (*Argos, 1990*; *Chen et al., 2013*), the artificial SNAP-25 linker was exclusively composed of glycine and serine residues (ratio of 2:1, varied sequence) in order to prevent interactions with the adjoining SNARE motifs (*Figure 1B*, cartoon). SN25 L$^{GGS}$ or WT SNAP-25 was expressed in KO chromaffin cells by means of a SF virus, and secretion was assayed by membrane capacitance measurements and amperometric recordings. While expression of WT SNAP-25 reinstated the full secretory response upon stimulation by Ca$^{2+}$-uncaging, no rescue of transmitter release was detected in KO cells expressing SN25 L$^{GGS}$ (*Figure 1Ba,b*). Reasoning that the linker mutant might have weakened stimulus-secretion coupling or otherwise delayed secretion, we performed membrane capacitance measurements for extended time intervals using strong Ca$^{2+}$-stimuli. Intriguingly, we found that even infusion of a 19 µM Ca$^{2+}$-containing solution via the patch pipette for up to 5 min did not result in a recovery of secretion in cells expressing SN25 L$^{GGS}$ (*Figure 1—figure supplement 2A*). Thus, the linker substitution mutant SN25 L$^{GGS}$ is dysfunctional, when expressed in KO cells. Nevertheless, SN25 L$^{GGS}$ exerted a mild dominant-negative effect in WT chromaffin cells, suggesting that the mutant still interacts with cognate SNARE partners via its intact SNARE motifs but fails to support exocytosis (*Figure 1—figure supplement 2B*). In order to visualize and quantify protein expression in our experiments, mutant and WT SNAP-25 carried an N-terminal GFP-tag, which was previously shown not to interfere with protein function (*Delgado-Martínez et al., 2007*). Since total GFP fluorescence intensity of cells expressing SN25 L$^{GGS}$ was indistinguishable from WT controls (*Figure 1—figure supplement 3A*), the observed secretion phenotype of SN25 L$^{GGS}$ cannot be simply due to insufficient expression of the mutant protein. Noteworthy, it has been reported before (*Nagy et al., 2008*) and is also demonstrated here (Figure 3A) that an acylation-deficient SNAP-25 mutant (SNAP-25 C$^{84}$S, C$^{85}$S, C$^{90}$S, C$^{92}$S; SN25 LN$^{4CS}$) can still mediate substantial amounts of release, which excludes the possibility that the loss of acylation and the resulting mislocalization of SN25 L$^{GGS}$ (*Figure 1—figure supplement 2C*) primarily account for the severe secretion deficits. Hence, our data suggest a mechanistic requirement for specific biophysical properties and/or special motifs of the SNAP-25 linker peptide during fusion.

## The SNAP-25 linker is critical for t-SNARE interaction and complex assembly

Substitution of the linker domain by an unrelated flexible peptide in SN25 L$^{GGS}$ may result in abnormal SNAP-25 folding and potentially could even affect the integrity of SNARE motifs. To investigate whether SN25 L$^{GGS}$ can still engage in ternary core complex formation, we mixed bacterially expressed SN25 L$^{GGS}$ or WT protein with syb-2 (GST-fusion protein) and stx-1A in equimolar concentrations (3 µM) and quantified the amount of assembled complexes by SDS-PAGE and Coomassie staining after defined time intervals (*Figure 2A*). Note that we did not work with an excess of SNAP-

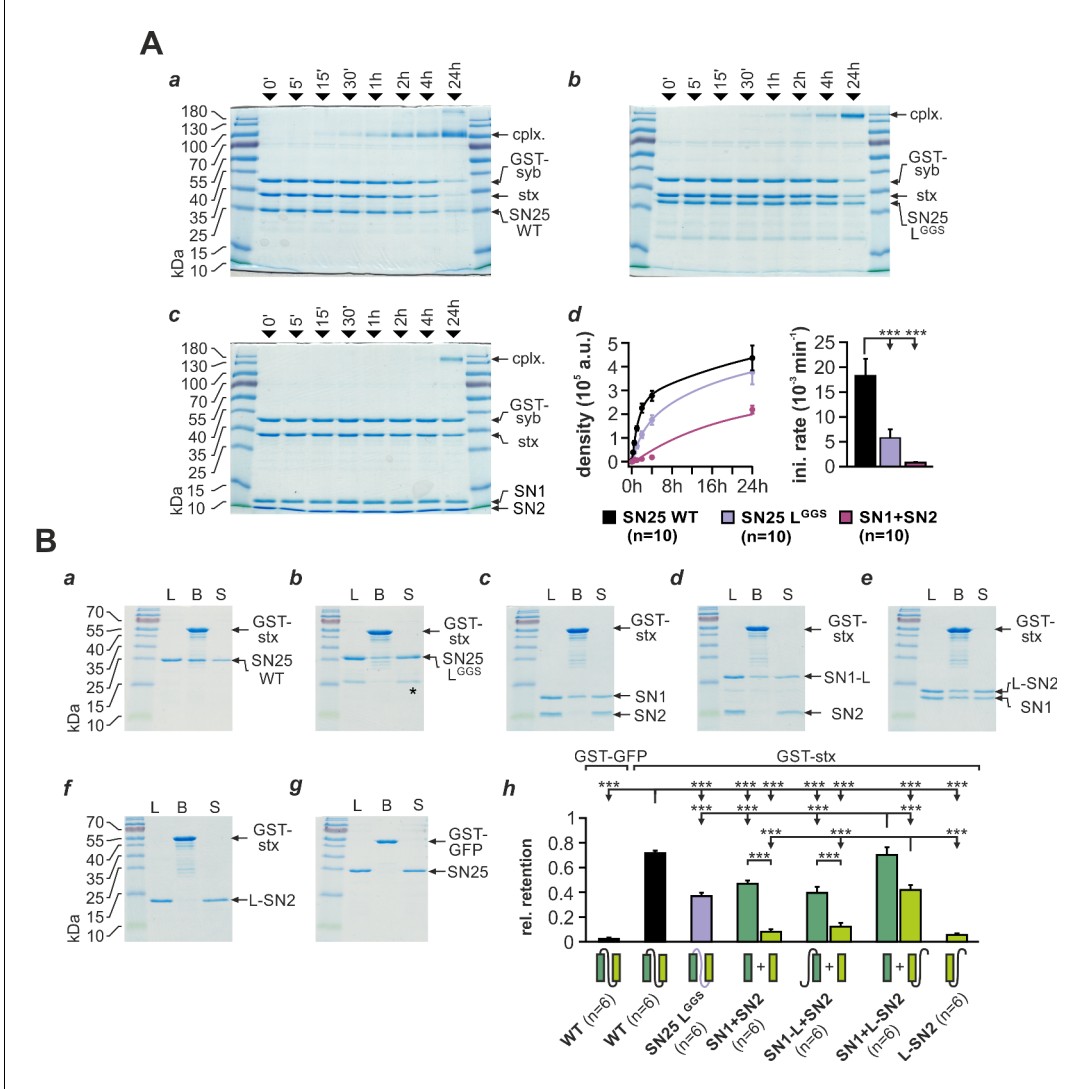

**Figure 2.** SNAP-25 linker motifs promote t-SNARE interactions and ternary complex assembly. (**A**) Kinetics of SNARE complex assembly was assayed by mixing equimolar amounts of purified GST-syb-2 (aa 1–116) and stx-1A (aa 1–262) with (**a**) SN25 WT, (**b**) SN25 L$^{GGS}$, or (**c**) isolated SN1/SN2 fragments. SNARE complex formation was analyzed by SDS PAGE after different times (indicated above lanes). (**d**) Quantitative analysis of complex formation. Biexponential fits of the kinetic profiles (*left panel*) demonstrated a significant decrease of the initial rate constant for SN25 L$^{GGS}$ and SN1/SN2 fragments (right panel). (**B**) Binding of SN25 L$^{GGS}$ or SNAP-25 fragments to immobilized stx-1A was tested in pulldown assays. Depicted are representative Coomassie-stained gels for different experimental conditions: (**a**) pulldown of SN25 WT, (**b**) SN25 L$^{GGS}$ ('*' marks a contaminating degradation product), (**c**) SN1 and SN2, (**d**) SN1-L and SN2, (**e**) SN1 and L-SN2, and (**f**) L-SN2. (**g**) GST-GFP was used as a negative control. Each gel shows load ('L'), GST-stx-1A containing beads after binding reaction ('B'), and supernatant ('S'). (**h**) To quantify the binding to immobilized stx-1A, we calculated the average retention ratio $n_{prey}/n_{stx-1A}$ for each protein/fragment. Depicted errors are SEM. *n* is indicated in the corresponding panels. Statistical analysis was done with ANOVA and Tukey's test.

DOI: https://doi.org/10.7554/eLife.41720.010

The following source data and figure supplements are available for figure 2:

**Source data 1.** Extended statistical data as Microsoft Excel spreadsheet.

DOI: https://doi.org/10.7554/eLife.41720.013

**Figure supplement 1.** A continuous L-SN2 fragments can facilitate ternary SNARE complex formation.

DOI: https://doi.org/10.7554/eLife.41720.011

**Figure supplement 1—source data 1.** Extended statistical data as Microsoft Excel spreadsheet.

DOI: https://doi.org/10.7554/eLife.41720.012

25 to suppress unproductive 2:1 stx-1A:SNAP-25 complexes (*Fasshauer and Margittai, 2004*), since this might have diminished potential kinetic differences between mutant and WT SNAP-25. As a 'negative' control, we also employed a 1:1 mixture of isolated SN1 (aa 1–82) and SN2 (aa 142–206) fragments.

Interestingly, our assay demonstrated that SN25 $L^{GGS}$ was still able to engage in SNARE complex formation under these conditions, which largely excludes a gross misfolding of the SNARE motifs (*Figure 2Ab*). The assembly kinetics of SNARE complexes containing SN25 $L^{GGS}$ was however clearly decelerated in comparison to WT controls (*Figure 2Ad*), which might be in part due to the high glycine content and the deviating biophysical properties of the artificial linker. Yet, complex formation in experiments with SN25 $L^{GGS}$ was considerably faster than the kinetics observed for separated SN1 and SN2 motifs (*Figure 2Ab-d*), emphasizing the importance of a physical linkage of both motifs for reaction speed. Fitting the kinetic profiles with a biexponential function, we found that the rate constant for the initial rapid phase of SNARE assembly was significantly decreased for SN25 $L^{GGS}$ in comparison to controls (*Figure 2Ad*). In case of linker-less SNAP-25 fragments, our analysis also revealed a strong decrease in the fast rate constant, which was accompanied by a reduction in the total amount of SNARE complexes assembled within 24 hr ($2.19*10^5 \pm 0.17*10^5$ a.u., WT: $4.37*10^5 \pm 0.52*10^5$ a.u.; p<0.01).

SNARE complexes formed by SN25 $L^{GGS}$ or separated SN1 and SN2 motifs exhibited a characteristic size shift (~180 kDa; *Figure 2Ab-c*), which possibly suggests the formation of SNARE complex dimers. Indeed, previous work by *Fdez et al. (2008)* has shown that two fully formed ternary complexes containing linker-less SN1/SN2 fragments can assemble into dimeric structures with attached C-terminal tips. Thus, the primary structure of the SNAP-25 linker peptide likely affects the biochemical properties of SNAREpins and their propensity to dimerize.

The SN25 $L^{GGS}$–induced slowdown of complex assembly may hint at a role of the linker in providing conformational support for SNARE complex nucleation. Therefore, we investigated whether linker substitution also affects the interplay of SNAP-25 and stx-1A. To evaluate linker:stx-1A interactions, we immobilized GST-stx-1A (aa 1–265) on glutathione-sepharose beads and quantified the 'pulldown' of SN25 $L^{GGS}$ or WT SNAP-25 (*Figure 2B*). Stx-1A-loaded beads were incubated with each prey protein (5 µM) for 20 hr at 4°C in order to approach binding equilibrium. Intriguingly, we found that SN25 $L^{GGS}$ bound to stx-1A at a ~ 50% lower molar ratio than WT protein (*Figure 2Ba,b, h*). Earlier biochemical work demonstrated that stx-1A only interacts with truncated SNAP-25 fragments that contain the unperturbed SN1 motif (*Chapman et al., 1994*), derogating the role of the linker and SN2 motif in directly binding to stx-1A H3. Given the low retention of SN25 $L^{GGS}$, we wondered whether linker motifs assist the recruitment of SN2 into helical assemblies, thereby intensifying stx-1A interactions. Thus, we tested whether inclusion of the linker in SNAP-25 fragments could enable pulldown of SN2. When an equimolar mix of isolated SN1 and SN2 fragments was incubated with stx-1A, SN1 was almost exclusively retained on beads (*Figure 2Bc,h*), as expected (*Chapman et al., 1994*). Adding the linker back to SN1 (aa 1–142; SN1-L) did not increase fragment binding, nor did it facilitate the retention of SN2 (*Figure 2Bd,h*). In contrast, a continuous linker-SN2 fragment (aa 83–206; L-SN2) bound to immobilized GST-stx-1A in the presence of isolated SN1 motifs (*Figure 2Be,h*) but not in their absence (*Figure 2Bf,h*). The amount of retained SN1 also significantly increased in the presence of L-SN2, suggesting that linker-mediated recruitment of SN2 also stabilizes the whole assembly. Thus, SN2 and linker form a functional unit that likely binds to SN1:stx-1A complexes.

As the requirement for a continuous linker-SN2 segment in t-SNARE interactions might limit ternary complex nucleation, we also compared SNARE complex assembly in mixtures containing either SN1-L and SN2, SN1 and L-SN2, or linker-less SN1 and SN2 together with stx-1A and GST-syb-2 (*Figure 2—figure supplement 1A*). We observed that the presence of the linker in either fragment combination significantly facilitated SNARE assembly over linker-less fragments, as judged by the time constants of monoexponential fits of assembly profiles (*Figure 2—figure supplement 1B*). Intriguingly, the combination of SN1/L-SN2-fragments was slightly more efficient than SN1-L/SN2 in promoting complex formation, indicating that linker interactions during initial t-SNARE assembly might indeed determine overall assembly kinetics. That said, our biochemical assays likely underestimate the effect of t-SNARE interactions, as fragments can easily interact with each other in solution without steric hindrance.

In summary, motifs within the native SNAP-25 linker are required for normal t-SNARE interactions as well as the fast assembly of ternary SNARE complexes. Accordingly, the severe secretory phenotype of SN25 L$^{GGS}$ is probably caused by ineffective SNARE interactions that hinder complex initiation.

## N- and C-terminal linker regions mechanistically support Ca$^{2+}$-dependent release

To learn more about the fusion-promoting function of the linker, we generated two partial substitution mutants, in which the N-terminal region (aa 83–118; SN25 LN$^{GGS}$) or the C-terminal linker portion (aa 119–141; SN25 LC$^{GGS}$) was exchanged by a corresponding flexible G/S peptide of equal length (*Figure 3A,B*). We based this subdivision on earlier reports that the N-terminal linker segment could be proteolytically excised (cuts at aa 84/115/120/125) from fully assembled complexes (*Fasshauer et al., 1998a*) and that a C-terminal motif (aa 120–129) determines isoform-specific release properties (*Nagy et al., 2008*). Expression of either mutant in KO chromaffin cells produced only small secretory responses in uncaging experiments (*Figure 3Aa,Ba*). Yet, secretion levels for both mutants clearly exceeded residual release in KO cells, indicating a partial functionality of the mutant proteins.

Fitting capacitance traces by exponential functions typically delivers detailed information about the different secretory components, especially about release from the *readily releasable pool* (RRP) and *slowly releasable pool* (SRP) (*Voets et al., 1999*). In case of SN25 LN$^{GGS}$, a detailed kinetic analysis showed that secretion is not only severely decreased but also markedly slowed-down in comparison to controls (*Figure 3Ab*). In particular, we observed a complete loss of the RRP component, a profound reduction in SRP size, a substantially prolonged time constant for release of SRP vesicles, a moderate decrease in sustained release, and a delay in the onset of secretion (*Figure 3Ab*). Hence, substitution of the N-terminal linker segment generated a 'complex' phenotype, simultaneously altering several mechanistic aspects of the fusion process including priming, primed vesicle stability, and fusion triggering. Noteworthy, SN25 LN$^{GGS}$ lacks the palmitoylation motif and consequently exhibits a dramatically reduced plasma membrane targeting (*Figure 3—figure supplement 1A*) despite a normal overall expression (*Figure 1—figure supplement 3A*). In comparison, the palmitoylation-deficient mutant SN25 LN$^{4CS}$, which is also not targeted to the plasma membrane (*Figure 3—figure supplement 1A*), showed much milder but clearly related release deficits (*Figure 3Aa,b*), indicating that the loss of membrane association is not the sole cause of diminished transmitter release.

The phenotype of the C-terminal substitution mutant SN25 LC$^{GGS}$ appeared almost identical to the one observed for SN25 LN$^{GGS}$ (*Figure 3Ba*; see normalized plots in *Figure 3—figure supplement 1B*). Our kinetic analyses showed that SN25 LC$^{GGS}$ strongly reduced the size of both primed vesicle pools and simultaneously increased the time constants for RRP- and SRP-mediated release in comparison to controls (*Figure 3Bb*). Additionally, secretion onset was prolonged, and sustained release was decreased for SN25 LC$^{GGS}$ (*Figure 3Bb*). Thus, C-terminal linker substitution also caused defects in priming, maintenance of primed vesicles, and triggering – very much like SN25 LN$^{GGS}$.

As structural data suggested an interaction of nonpolar side-chains in the very C-terminal linker segment with a hydrophobic groove on the core SNARE complex (*Sutton et al., 1998*), we further characterized a mutant, in which the four corresponding amino acids were substituted by glycine residues (F$^{133}$G, I$^{134}$G, V$^{137}$G, A$^{141}$G; denoted SN25 LN$^{4G}$; *Figure 3B*). SN25 LC$^{4G}$ produced a substantially smaller and slower secretory response than in controls, as indicated by a strong reduction in RRP size and an increased time constant of RRP release (*Figure 3Bb*). Hence, even this small substitution induced a complex release phenotype that is reminiscent of the defects seen in SN25 LC$^{GGS}$. The phenotypic similarity to SN25 LC$^{GGS}$ became even stronger, when more residues in the very C-terminal linker segment were mutated, as demonstrated by replacing the last nine amino acids by glycine residue in SN25 LC$^{9G}$ (*Figure 3—figure supplement 2*). Clearly, the potential core complex interaction site and the adjoining linker sequence are important for a fusion intermediate that determines the fate of the primed vesicle. Note that all C-terminal linker mutants were normally targeted to the plasma membrane and exhibited a similar overall expression level like the WT protein in controls (*Figure 1—figure supplement 3A,Bb*), excluding the possibility that the observed secretion deficits are due to reduced protein availability.

Ternary SNARE complex assembly appeared only slightly decelerated in biochemical experiments with SN25 LN$^{GGS}$ and SN25 LC$^{GGS}$ (*Figure 3—figure supplement 1Ca–d*). Biexponential fits of the

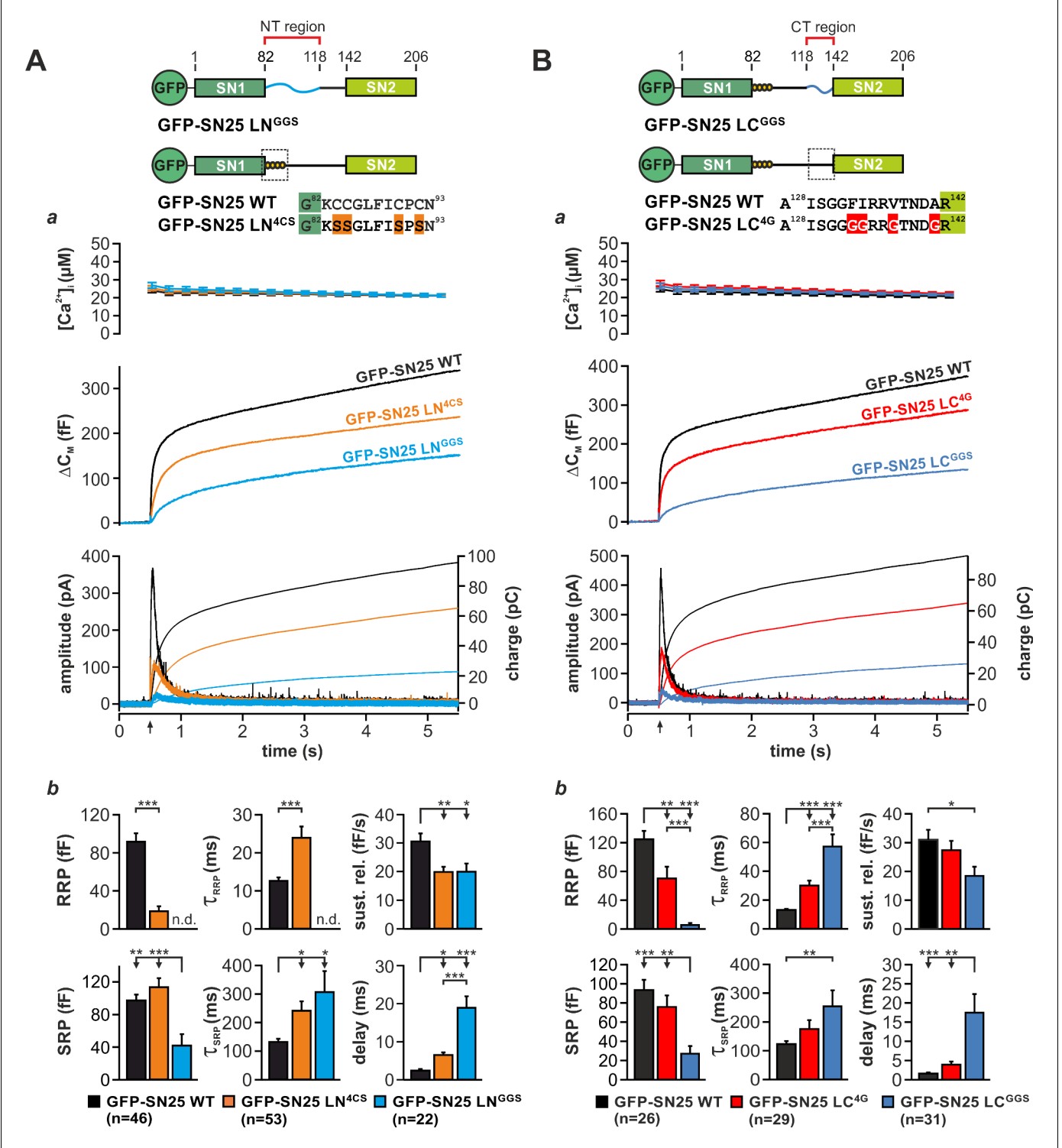

**Figure 3.** Motifs in N- and C-terminal linker regions promote secretion. (**A**) Characterization of secretion deficits induced by GFP-SN25 LN$^{GGS}$ or GFP-SN25 LN$^{4CS}$. Cartoon illustrates the structure of the tested mutants. (**a**) Averaged traces for [Ca$^{2+}$]$_i$ (*top panel*), capacitance measurements (*middle*) and amperometric recordings (*bottom*) in KO cells expressing GFP-SN25 LN$^{GGS}$ (cyan), GFP-SN25 LN$^{4CS}$ (orange), or WT protein (black) are shown. Arrow indicates UV flash. (**b**) Quantitative analysis of the kinetic properties of the secretory burst. The mean amplitude of RRP and SRP as well as the corresponding time constants $\tau_{fast}$ and $\tau_{slow}$ are depicted. Sustained release rate reflects the slope of the linear secretion component. The mean exocytotic delay indicates the average latency of secretion after the flash. (**B**) Characterization of secretion in KO cells expressing GFP-SN25 LC$^{GGS}$ or GFP-SN25 LC$^{4G}$. Cartoon illustrates the structure of the C-terminal linker substitutions mutants. (**a**) Averaged recordings for the GFP-SN25 LC$^{GGS}$

*Figure 3 continued on next page*

*Figure 3 continued*

mutant (blue), GFP-SN25 LC[4G] (red) or wildtype controls (black) are shown. Panel organization as in (Aa). (**b**) Quantitative analysis of kinetic parameters. All data are presented as mean ± SEM; *n* is indicated in the corresponding panels. Statistical comparisons were performed with ANOVA and Tukey's post-hoc test.

DOI: https://doi.org/10.7554/eLife.41720.014

The following source data and figure supplements are available for figure 3:

**Source data 1.** Extended statistical data as Microsoft Excel spreadsheet.
DOI: https://doi.org/10.7554/eLife.41720.019
**Figure supplement 1.** Extended characterization of the partial linker substitution mutants SN25 LN[GGS] and SN25 LC[GGS].
DOI: https://doi.org/10.7554/eLife.41720.015
**Figure supplement 1—source data 1.** Extended statistical data as Microsoft Excel spreadsheet.
DOI: https://doi.org/10.7554/eLife.41720.016
**Figure supplement 2.** Complex secretion phenotype of an extended substitution mutant targeting the backfolding C-terminal linker loop.
DOI: https://doi.org/10.7554/eLife.41720.017
**Figure supplement 2—source data 1.** Extended statistical data as Microsoft Excel spreadsheet.
DOI: https://doi.org/10.7554/eLife.41720.018

kinetic profiles still revealed a significantly decreased initial rate constant for SN25 LN[GGS] and SN25 LC[GGS] in comparison to controls (*Figure 3—figure supplement 1Ce,f*). Thus, N- and C-terminal linker regions each must contain motifs that provide additive support for SNARE assembly. Noteworthy, SNARE complexes formed by SN25 LN[GGS] (but not SN25 LC[GGS]) also exhibited a characteristic shift towards higher molecular mass in this assay. Furthermore, we tested both linker substitution mutants in pulldown experiments with immobilized stx-1A (*Figure 3—figure supplement 1D*) and found that exclusively the retention of SN25 LC[GGS] was significantly reduced compared to WT SNAP-25 (*Figure 1—figure supplement 1Dd*). This finding highlights that only the C-terminal linker motif substantially stabilizes t-SNARE dimers, while the contribution of the N-terminal linker region is negligible.

In summary, our data suggest that motifs in the N-terminal as well as in the C-terminal linker segment are critical for fast, $Ca^{2+}$-dependent vesicle fusion. Both linker segments likely stabilize late fusion intermediates in a cooperative fashion, preventing unpriming and promoting $Ca^{2+}$-dependent fusion triggering.

## The acylated N-terminal linker segment promotes triggering

While priming-related release deficits of linker substitution mutants can be readily explained by perturbed SNARE assembly, the observed kinetic slow-down of secretion also suggests a distinctive function of linker motifs in late fusion steps. As mutation of the four acylated cysteine residues in the N-terminal linker segment (SN25 LN[4CS]) suffices to decelerate release kinetics and decrease apparent RRP size (*Figure 3Aa,b*; cf. *Nagy et al., 2008*), we wondered about a specific mechanistic contribution of the acylated linker in fusion triggering. Interestingly, the palmitoylation motif is located directly downstream of SN1 in all known orthologs of SNAP-25, thus mediating membrane contacts near the C-terminal end of the assembling SNARE complex. To distinguish the effects of local lipid interactions from potential deficits due to a general protein mislocalisation, we attached the minimal palmitoylation motif of SNAP-25 (aa 83–120) (*Gonzalo et al., 1999*) to the N-terminus of GFP-tagged SN25 LN[4CS] (denoted P-GFP-SN25 LN[4CS]), thereby providing a secondary acylation point for re-targeting to the plasma membrane (*Figure 4A*, cartoon). Indeed, P-GFP-SN25 LN[4CS] was almost exclusively localized at the plasma membrane of infected KO cells (*Figure 4A*). WT SNAP-25 carrying the same modification (P-GFP-SN25 WT) showed a similar membrane expression, only slightly deviating from the distribution pattern of P-GFP-SN25 LN[4CS] by a stronger intracellular protein accumulation (*Figure 4A*, example image), which likely represents a Golgi/recycling endosomal SNAP-25 pool (*Aikawa et al., 2006*). P-GFP-SN25 WT efficiently rescued secretion in KO chromaffin cells and even reached a slightly higher release level than typically seen in other control recordings (*Figure 4B*). This increase in total secretion might be attributed to a more pronounced membrane targeting compared to the unmodified WT protein. In contrast, the retargeted P-GFP-SN25 LN[4CS] mutant still caused secretion deficits similar to the original acylation mutant despite its high membrane expression (*Figure 4Ba,b*; cf. *Figure 3Aa,b*). Restoration of membrane localization clearly did

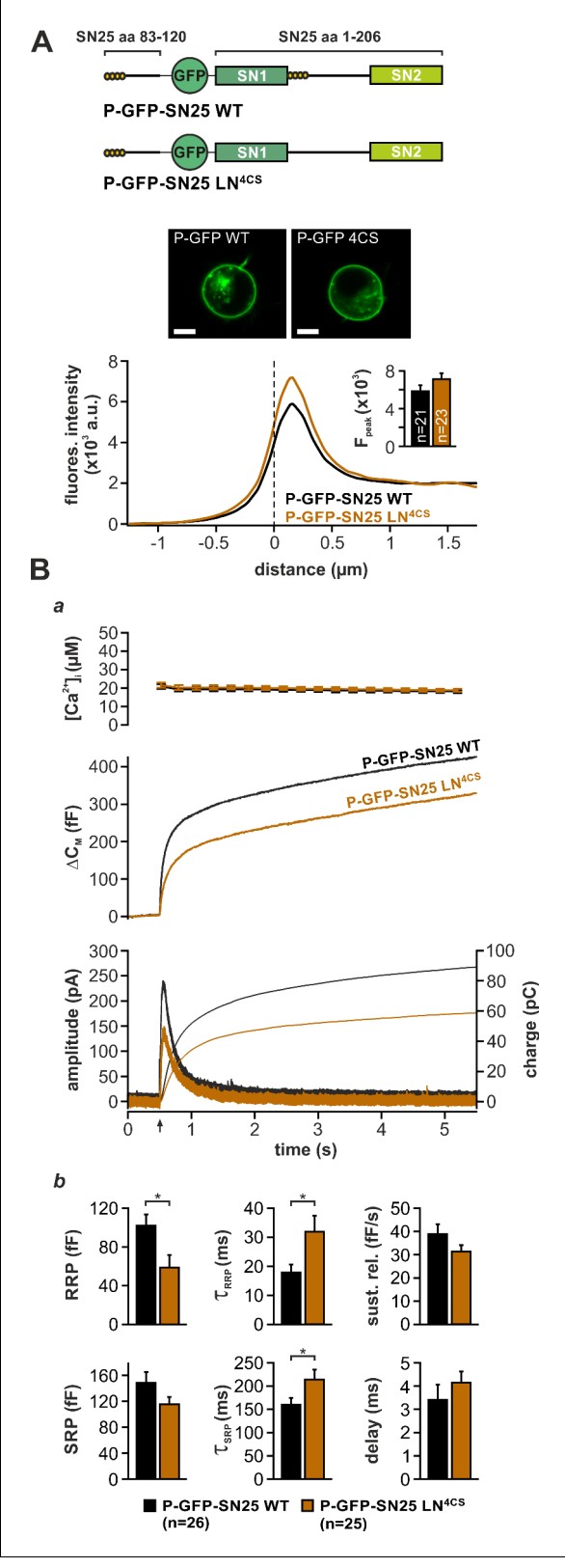

**Figure 4.** Intramolecular acylation of SNAP-25 is required for efficient triggering and stability of primed vesicles. (**A**) Cartoon illustrates the construction of SNAP-25 variants carrying a secondary palmitoylation motif fused to the GFP-tag. The cellular localization of SNAP-25 variants was analyzed by confocal microscopy. Exemplary pictures of KO cells expressing P-GFP-SN25 LN[4CS] or double-acylated WT protein are shown. Scale bar is 5 μm. Mean radial
*Figure 4 continued on next page*

*Figure 4 continued*

line scans for P-GFP-SN25 LN$^{4CS}$ (red) and P-GFP-SN25 WT (black) demonstrate similar plasma membrane localization. Inset depicts the average fluorescence near the plasma membrane. (**B**) Characterization of secretion properties in KO cells expressing P-GFP-SN25 LN$^{4CS}$ (red) or P-GFP-SN25 WT (black). (**a**) Shown are averaged Ca$^{2+}$-imaging results (*top*), capacitance measurements (*middle*) and amperometric recordings (*bottom*). Arrow indicates UV flash. (**b**) Quantitative analysis of RRP and SRP size, time constants of release components, sustained release rate, and exocytotic delay. Data are depicted as mean ± SEM; *n* is indicated in the panels. Statistical comparisons were done using Student's t-test.

DOI: https://doi.org/10.7554/eLife.41720.020

The following source data is available for figure 4:

**Source data 1.** Extended statistical data as Microsoft Excel spreadsheet.

DOI: https://doi.org/10.7554/eLife.41720.021

not abate the primary deficits in fusion triggering and pool stability. Hence, we conclude that linker acylation at a defined intramolecular position is required for efficient membrane fusion, pointing to a mechanistic role for local linker-mediated membrane contacts.

In a plausible scenario, the linker-based membrane anchor might facilitate fusion by relaying SNARE-generated force onto the plasma membrane. To further test this idea, we constructed linker mutants, in which the distance between the acylation site and SN1 was increased by insertion of flexible spacer peptides (6, 10, or 14 residues; mutants denoted SN25 shift6aa/10aa/14aaL) in order to mechanically 'uncouple' the acyl anchors from the core complex (*Figure 5A*). We especially relied on G/S-containing spacers, as the high rotational freedom of the included glycine residues should substantially lessen conformational strain and thereby further hamper force transfer. Deviating from the secretion phenotype of SN25 LN$^{4CS}$, none of these spacer insertions caused a detectable reduction in the size of the secretory burst (exemplified by SN25 shift14aaL in *Figure 5Ba*). However, all mutants consistently induced a moderate slow-down of release, as seen in scaled plots of $\Delta C_M$ traces (*Figure 5Bc*) and reflected by increased time constants of RRP and SRP-mediated secretion in the kinetic analysis (*Figure 5Bb*). Moreover, we observed a significantly prolonged latency of the secretory burst in all mutants that further confirmed deficits in fusion triggering (*Figure 5Bb*). Since total expression and membrane localization of the three mutants were indistinguishable from WT controls (*Figure 1—figure supplement 3A,Bc*), the kinetic changes cannot be due to a shortage of mutant protein on the plasma membrane. Intriguingly, the phenotype is also not caused by increased linker length per se, as the initial linker segment can be extended by seven amino acids without functional deficits using natural insertions found in the longer N-terminal linker region of the related SNAP-23 isoform (*Figure 5—figure supplement 1A,B*). Similarly, a net extension of the linker by 14 amino acids using an additional spacer after A$^{118}$ was well tolerated (*Figure 5—figure supplement 1A,C*).

In order to directly test the idea of linker-mediated force transfer, we next introduced the longest spacer peptide into SN25 LN$^{4CS}$, generating a double mutation denoted SN25 shift14aaLN$^{4CS}$ (*Figure 5A*). The rationale behind this experiment was that elimination of SNAP-25 acylation should break up the most relevant mechanical coupling between linker and membrane, as the binding free energy of four lipid-embedded palmitoyl groups (*Pool and Thompson, 1998*) should by far exceed the energetic contributions of other linker:membrane interactions. In comparison, the free energy gain for electrostatic membrane interactions of lysine residues (five in the N-terminal linker) is very small (*Kim et al., 1991*), and unstructured peptide strands are generally expected to exhibit a low binding affinity to the membrane interface (*Almeida, 2014*; *White and Wimley, 1999*). Without a significant mechanical coupling between linker and plasma membrane, no exacerbation of the phenotype would be expected for SN25 shift14aaLN$^{4CS}$ in comparison to SN25 LN$^{4CS}$. Contrary to our expectation, however, the secretory burst in SN25 shift14aaLN$^{4CS}$ –expressing cells was further decelerated (*Figure 5Ca*), as can be well appreciated in normalized plots of $\Delta C_M$ (*Figure 5Cc*) and is evident from a significant decrease in RRP/SRP fusion rates and the prolonged exocytotic delay in kinetic analyses (*Figure 5Cb*). The substantially aggravated phenotype of SN25 shift14aaLN$^{4CS}$ in combination with the generally mild phenotypes of uncoupling spacer insertion mutants clearly argue against a prominent contribution of the linker to SNARE force transfer. Still, given the pronounced kinetic phenotype of the non-palmitoylated SN25 LN$^{4CS}$ mutant and its retargeted P-GFP-SN25 LN$^{4CS}$ variant, the membrane interactions of the acyl anchors near the C-terminal end

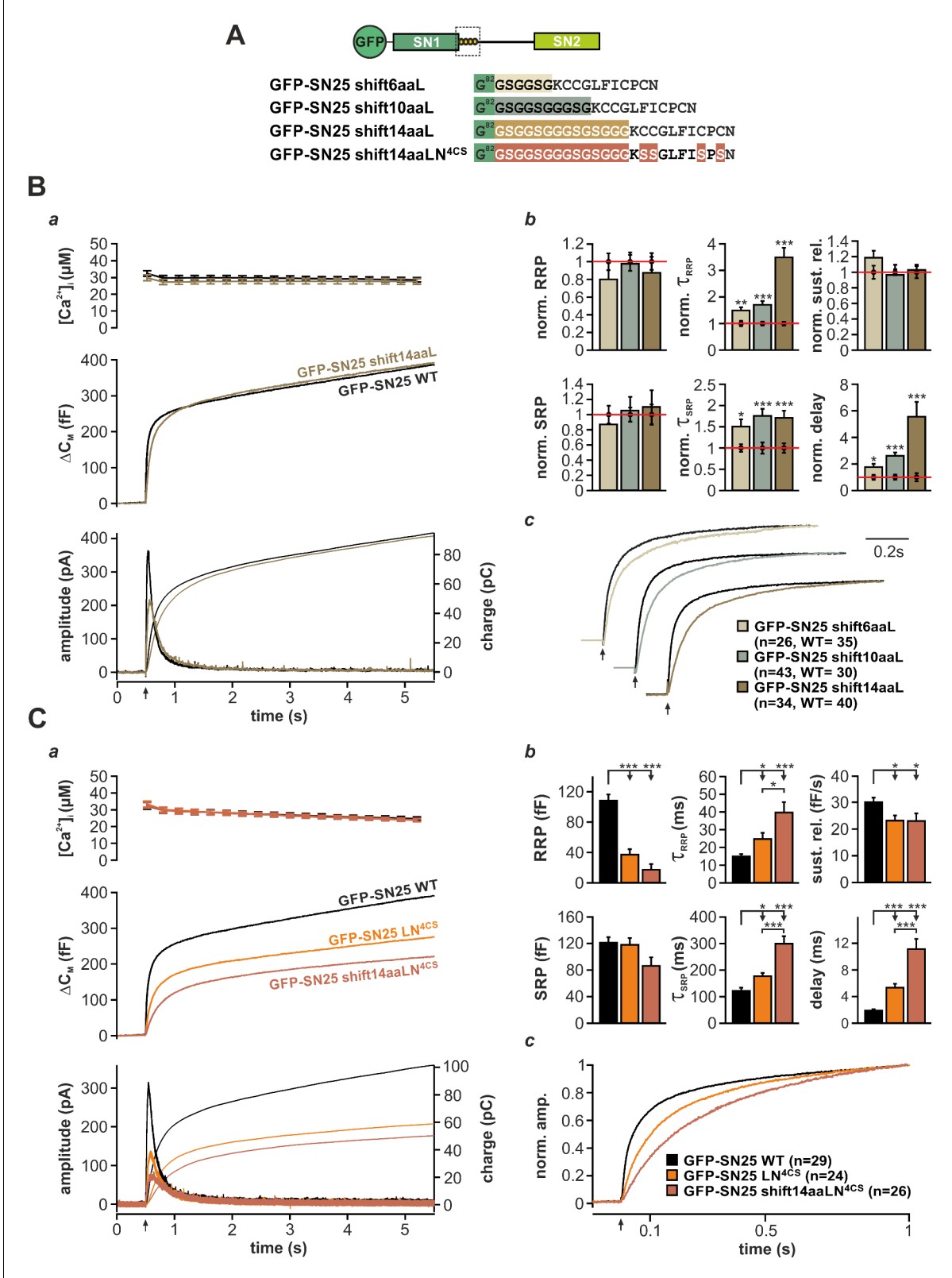

**Figure 5.** The acylated N-terminal linker motif does not play an essential role in force transfer. (**A**) Structure of the three mutants SN25 shift6/10/14aaL, which contain 'uncoupling' spacers, and the double mutant SN25 shift14aaLN⁴ᶜˢ. (**B**) The secretion phenotype of GFP-SN25 shift14aaL (olive; WT in black) is shown as example for the functional deficits induced by spacer insertions. (**a**) Averaged $[Ca^{2+}]_i$ measurements (*top*), capacitance traces (*middle*) and amperometric recordings (*bottom*) are shown. UV flash is indicated by arrow. (**b**) Analysis of kinetic parameters for all three tested mutants: GFP-

*Figure 5 continued*

SN25 shift6aaL (beige), GFP-SN25 shift10aaL (gray), and GFP-SN25 shift14aaL (olive). Data for the kinetic parameters were normalized to the corresponding WT control values. Statistical analyses were performed between mutant and control measurements (Student's t-test). (c) Normalized plots of averaged capacitance traces highlight the kinetic deceleration caused by spacer insertions. All three pairs of WT/mutant traces are equally scaled. (C) Comparative electrophysiological analysis of secretion in GFP-SN25 LN[4CS] (orange), GFP-SN25 shift14aaLN[4CS] (red-brown), and WT protein-expressing KO cells (black). (a) Panel organization as in (Ba). (b) Quantitative analysis of kinetic parameters. (c) Normalized plot of the secretory burst component highlighting the changed kinetics of the double mutant. All data are mean ± SEM; *n* is indicated in the corresponding panel. Statistical comparisons were performed using ANOVA and Tukey's test.

DOI: https://doi.org/10.7554/eLife.41720.022

The following source data and figure supplements are available for figure 5:

**Source data 1.** Extended statistical data as Microsoft Excel spreadsheet.

DOI: https://doi.org/10.7554/eLife.41720.027

**Figure supplement 1.** Internal extension of the linker does not alter secretion properties.

DOI: https://doi.org/10.7554/eLife.41720.023

**Figure supplement 1—source data 1.** Extended statistical data as Microsoft Excel spreadsheet.

DOI: https://doi.org/10.7554/eLife.41720.024

**Figure supplement 2.** Removal of lysine residues in vicinity of the palmitoylation motif slightly decreases release rate.

DOI: https://doi.org/10.7554/eLife.41720.025

**Figure supplement 2—source data 1.** Extended statistical data as Microsoft Excel spreadsheet.

DOI: https://doi.org/10.7554/eLife.41720.026

of the core complex must be mechanistically important for efficient fusion initiation and likely promote a favorable membrane configuration for merger.

The acerbated release deficits of SN25 shift14aaLN[4CS] also highlight acylation-independent functions of the N-terminal linker in fusion triggering, possibly involving additional lipid or protein interactions. Work by *Weber et al. (2017)* recently suggested that the N-terminal linker motif mediates transient membrane contacts via electrostatic interactions between lysine side-chains and phospholipid head groups. To test whether these weak lipid interactions are functionally important for secretion, we mutated five lysine residues in the N-terminal linker region of SN25 LN[4CS] ($K^{83}A$, $K^{94}A$, $K^{96}A$, $K^{102}A$, $K^{103}A$; denoted SN25 LN[5KA,4CS]; *Figure 5—figure supplement 2A*, cartoon), and analyzed potential effects on secretion in $Ca^{2+}$-uncaging experiments. Substitution of the lysine residues very slightly decelerated secretion in comparison to SN25 LN[4CS], as indicated by an increased time constant of RRP release (*Figure 5—figure supplement 2A–C*). Since no difference in overall expression or membrane targeting of SN25 LN[5KA,4CS] compared to SN25 LN[4CS] was noticeable (*Figure 1—figure supplement 3A,Ba*), we conclude that the lysine residues contribute to the linker:lipid interactions that facilitate $Ca^{2+}$-dependent fusion triggering. Noteworthy, the role of the lysine residues in the linker:membrane interface is probably more pronounced within the intact acylated linker but would be difficult to study due to an involvement in the acylation process (*Weber et al., 2017*).

In sum, our data suggest that the N-terminal linker region engages in different interactions with the plasma membrane that are important for late mechanistic steps of the fusion process. While these linker:membrane contacts may only play a limited role in force transfer, our results hint at a mechanistic scenario, in which acyl anchors and local electrostatic membrane interactions cooperatively establish a suitable membrane configuration for fusion triggering.

## The N-terminal linker section controls fusion pore expansion

To investigate whether the linker is also required for normal fusion pore evolution, we elicited secretion by infusion of an intracellular solution containing 19 µM $Ca^{2+}$ and studied the kinetics of amperometric spikes in KO cells expressing different linker mutants (*Figure 6*). In accord with *Nagy et al., 2008*, we found that SN25 LN[4CS] significantly prolonged the pre-spike signal (PS) (*Figure 6C*) but left the main spike signal unperturbed (*Figure 6B*). Interestingly, the N-terminal substitution mutant SN25 LN[GGS] affected the PS as well as the kinetics of the main spike: Similar to SN25 LN[4CS], the PS was significantly prolonged (*Figure 6Cc*), exhibiting reduced amplitudes (*Figure 6Ca,b*) and diminished current fluctuations (*Figure 6Ce,f*). Additionally, rise time and half width of the main spike signal were significantly increased (*Figure 6Bc,d*), while the total charge remained unchanged (*Figure 6Ba*). The effects on main spike waveform induced by SN25 LN[GGS] clearly indicate that the

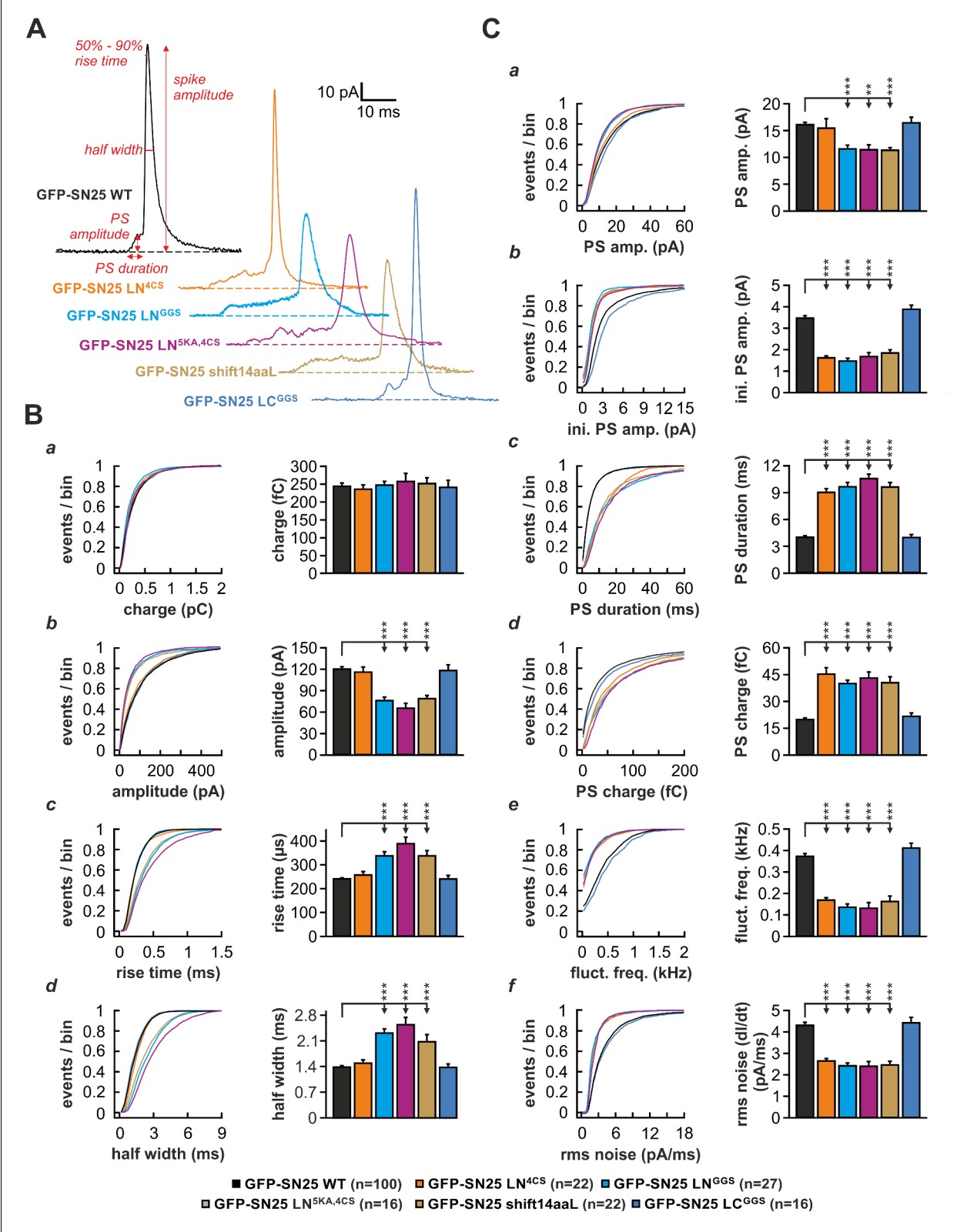

**Figure 6.** The N-terminal linker segment controls fusion pore expansion. (**A**) Exemplary amperometric spikes recorded in KO cells expressing different N- and C-terminal linker mutants. The kinetic parameters determined during analysis are marked for a spike recorded in control experiments with WT SNAP-25. (**B**) Quantitative analysis of main spike properties. Data were collected for the following conditions (# of events/ # of cells): WT protein (black, 12633/100), GFP-SN25 LN[4CS] (orange, 1419/22), GFP-SN25 LN[GGS] (cyan, 1986/27), GFP-SN25 LN[5KA,4CS] (grey, 1391/16), GFP-SN25 shift14aaL (olive,

*Figure 6 continued on next page*

*Figure 6 continued*

2377/22) and GFP-SN25 LC$^{GGS}$ (blue, 1014/16). Cumulative frequency distributions are displayed for (a) spike charge, (b) amplitude, (c) 50–90% rise time, and (d) half width. Bar graphs depict means of the median determined from the frequency distribution for each cell. (C) Analysis of PS waveform. Cumulative frequency distributions for (a) PS amplitude, (b) initial amplitude, (c) duration, (d) charge, (e) fluctuation frequency, and (f) root mean square (rms) noise of the current derivative are shown. As in (B), bar graphs show mean of cell medians for each parameter. Errors are SEM. Statistical comparisons were performed by ANOVA using Tukey's test.

DOI: https://doi.org/10.7554/eLife.41720.028

The following source data is available for figure 6:

**Source data 1.** Extended statistical data as Microsoft Excel spreadsheet.

DOI: https://doi.org/10.7554/eLife.41720.029

N-terminal linker region surrounding the cysteine cluster participates in the control of late fusion pore expansion. Confirming this notion, removal of the flanking lysine residues in SN25 LN$^{5KA,4CS}$ was sufficient to delay main spike kinetics, establishing a congruent phenotype to SN25 LN$^{GGS}$ (*Figure 6B–C*). Moreover, similar deficits in early and late fusion pore behavior were observed for SN25 shift14aaL (*Figure 6B–C*), demonstrating that a displacement of the palmitoylation motif effectively disturbs the linker-mediated support of pore dilation. The lack of kinetic alterations in cells expressing the C-terminal substitution mutant SN25 LC$^{GGS}$ (*Figure 6B–C*) finally suggests that fusion pore regulation is selectively tied to the N-terminal linker region. Hence, our data suggest that lipid interplay of the N-terminal linker segment with the neck of the fusion pore governs the behavior of the pore from its initial opening to the final stages of membrane merger.

## Rescue of linker-induced secretion deficits by membrane-active agents

If N-terminal linker:membrane interactions critically modulate lipidic fusion intermediates, application of membrane-active reagents might be able to 'rescue' release deficits. The underlying idea is that linker:membrane interplay might mechanistically help to facilitate spontaneous lipid stalk formation and membrane merger, and thus might serve a function that could be mimicked by application of membrane-perturbing compounds. While such treatments generally promote fusion (in mutants as well as in controls), the relative effect should be disproportionally strong in KO cells expressing SN25 LN$^{GGS}$, if the action of the compound can lessen the impact of the mechanistic 'bottleneck' imposed by the mutation.

Application of short-chain alcohols was reported to facilitate hemifusion/fusion of liposomes in reduced protein-free systems by altering membrane properties (*Chanturiya et al., 1999*; *Paxman et al., 2017*). Here, we introduced low concentrations (2%) of methanol (MeOH) into cells via the patch pipette and characterized potential changes of secretion properties in Ca$^{2+}$-uncaging experiments. Indeed, MeOH infusion into SNAP-25 WT-expressing KO cells mildly increased total secretion (+25%) by strengthening the SRP component (*Figure 7*). The very same treatment resulted in a profound recovery of the secretory burst in SN25 LN$^{GGS}$–expressing KO cells, basically doubling total secretion (*Figure 7B*). The restoration of release suggests that the alcohol-induced changes in lipid properties can partially compensate for the loss of linker:membrane interactions. Interestingly, MeOH treatment of SN25 LN$^{GGS}$–expressing KO cells selectively reconstituted primed vesicle pools and the sustained release component, leaving release rates unchanged (*Figure 7C*). Thus, our data primarily point to a MeOH-induced recovery of priming and priming stability. As short-chain alcohols likely affect hydration repulsion (*Ly and Longo, 2004*; *Paxman et al., 2017*), it stands to reason that MeOH compensates for an insufficient local shielding of the plasma membrane surface caused by elimination of local linker:lipid interactions.

Curvature-inducing lipids are known to either facilitate or inhibit membrane merger in the context of various model systems, including liposome fusion and virus-mediated membrane fusion (*Chernomordik and Kozlov, 2003*; *Melia et al., 2006*). As it was reported that cone-shaped lipids support the highly-bent stalk intermediate during fast Ca$^{2+}$-triggered fusion of secretory vesicles (*Churchward et al., 2008*), we investigated whether induction of negative curvature by infusion of oleic acid (OA) can abate the phenotype of SN25 LN$^{GGS}$. In control experiments, we infused SNAP-25 WT–expressing KO cells with intracellular solution containing 19 µM Ca$^{2+}$ and 5 µM OA and characterized secretion by membrane capacitance measurements as well as amperometric recordings.

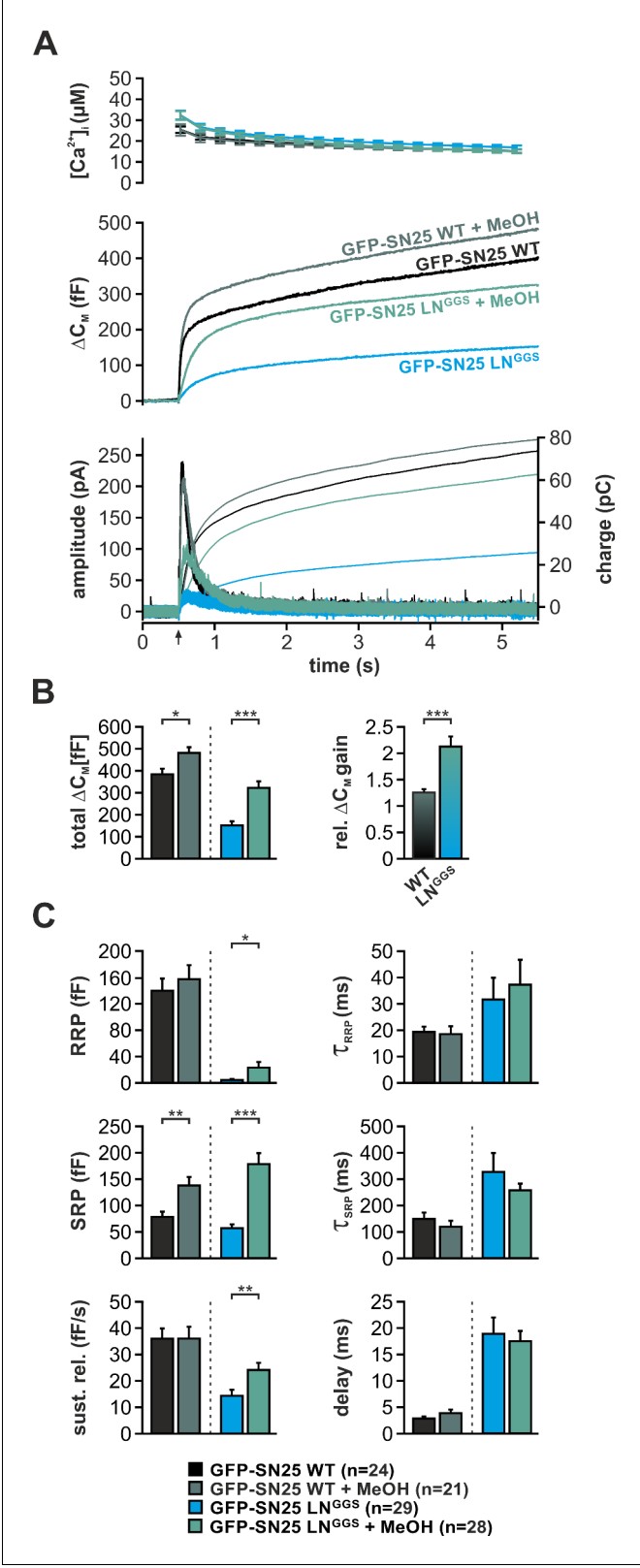

**Figure 7.** Intracellular application of methanol (MeOH) compensates for release deficits of GFP-SN25 LN$^{GGS}$. (**A**) The effect of low amounts of MeOH (2%) on secretion was tested in KO cells expressing either GFP-SN25 WT (dark green; untreated: black) or GFP-SN25 LN$^{GGS}$ (bright green; untreated: cyan) using Ca$^{2+}$-uncaging. Averaged [Ca$^{2+}$]$_i$ measurements (*top*), capacitance traces (*middle*) and amperometric recordings (*bottom*) are shown. UV

*Figure 7 continued*

flash is indicated by arrow. (B) Total $\Delta C_M$ was quantified 5 s after flash application (*left*) and the relative gain in secretion in the presence of MeOH was calculated. Note the significantly stronger effect of MeOH on GFP-SN25 LN$^{GGS}$-mediated release. (C) Analysis of kinetic parameters for secretion in presence and absence of MeOH. RRP and SRP size, time constants of release components, sustained release rate, and exocytotic delay were quantified. Data are shown as mean ± SEM; *n* is indicated at the bottom. Statistical comparisons between treated and untreated conditions were done using Student's t-test.

DOI: https://doi.org/10.7554/eLife.41720.030

The following source data is available for figure 7:

**Source data 1.** Extended statistical data as Microsoft Excel spreadsheet.
DOI: https://doi.org/10.7554/eLife.41720.031

OA infusion in these control experiments induced a ~ 50% gain in total capacitance over 2 min compared to vehicle-treated cells (*Figure 8A*). In stark contrast, infusion of SN25 LN$^{GGS}$–expressing KO with OA almost tripled the membrane capacitance increase seen within 2 min (*Figure 8A*), indicating that the facilitating action of OA can in part accommodate for the loss of N-terminal linker interactions. Hence, we propose that the membrane anchor and flanking linker sequences support a critical lipidic fusion intermediate that is sensitive to curvature-inducing agents.

As curvature-perturbing lipids have been shown to affect the life-time of the nascent fusion pore (*Zhang and Jackson, 2010*), we also investigated whether OA application could reverse the kinetic slow-down of amperometric spikes in SN25 LN$^{GGS}$-expressing cells (*Figure 8B,C*). In control experiments with SNAP-25 WT-expressing cells, application of OA resulted in a clear acceleration of PS and main spike waveform, as reflected by a decreased rise time and half width (*Figure 8Bc,d*), an increased spike amplitude (*Figure 8Bb*), and a shortening of PSs (*Figure 8Ca*). A similar kinetic speed-up of spike kinetics was also observed, when OA was introduced into SN25 LN$^{GGS}$–expressing KO cells (*Figure 8B,C*), in result compensating for all decelerating effects of the linker mutation on PS and main spike signals. The full reversal of the kinetic deficits of SN25 LN$^{GGS}$ suggests that linker-mediated membrane interactions and OA both influence the same mechanistic determinant of pore evolution. Thus, our data suggest the possibility that the N-terminal lipid contacts could directly regulate pore expansion by altering membrane curvature near the fusion pore neck.

## Discussion

Despite the fundamental mechanistic role of SNARE proteins in fast $Ca^{2+}$-triggered exocytosis, the functional implications of the tandem-like structural organization of SNAP-25a have not been sufficiently understood. Here, we studied the mechanistic role of the linker domain of SNAP-25, uncovering two unrecognized mechanistic functions: [1] Linker motifs govern SNARE protein interactions and facilitate the formation of binary and ternary complexes. [2] The N-terminal linker segment engages in local membrane contacts that support critical lipidic fusion intermediates.

### The SNAP-25 Linker is a functional component of the fusion machinery

Earlier reconstitution experiments with two complementing SNAP-25 fragments indicated that a physical linkage of SNARE domains in SNAP-25 is not a prerequisite of $Ca^{2+}$-dependent exocytosis (*Chen et al., 1999*; *Wang et al., 2008*). While we confirmed here that co-expression of separate SN1- and SN2-containing fragments was in principle sufficient to reinstate $Ca^{2+}$-dependent vesicle fusion in KO chromaffin cells, an elaborated analysis showed that secretory responses were dramatically diminished and kinetically distorted in spite of an abundance of expressed fragments (*Figure 1A*, *Figure 1—figure supplement 1B*). Hence, our results suggest that the structural integrity of the linker domain is indeed important for fast $Ca^{2+}$-triggered exocytosis in neuroendocrine cells, refuting the prevalent notion of a functionally dispensable linker domain. Previous work might have underestimated the relevance of the linker due to technical limitations of the employed BoTn / E based rescue paradigms and population-based secretion assays (*Chen et al., 1999*). Moreover, secretion in permeabilized or cracked cells might not follow canonic release pathways (*Wang et al., 2008*). To remedy such experimental shortcomings, we have performed all rescue experiments in

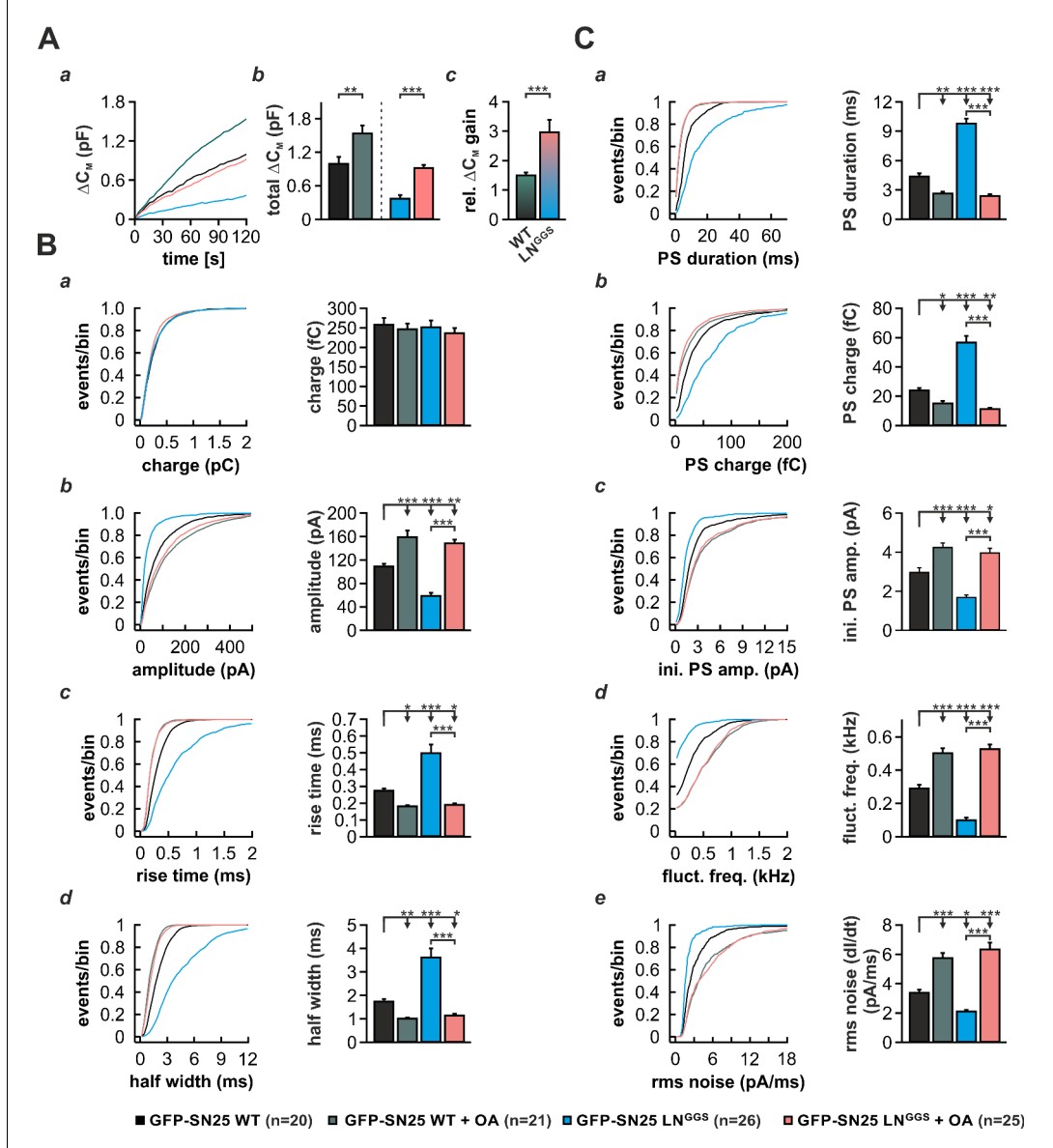

**Figure 8.** Intracellular infusion of oleic acid (OA) partially rescues exocytosis and reverses the kinetic delay of amperometric spikes induced by GFP-SN25 LN$^{GGS}$. (**A**) Characterization of secretion from KO cells expressing SN25 LN$^{GGS}$ or WT protein during infusion of solution containing 19 µM free Ca$^{2+}$ together with either OA (5 µM) or vehicle (DMSO). (**a**) Depicted is the mean capacitance change ($\Delta C_M$) over 120 s for each condition. (**b**) Total $\Delta C_M$ after 120 s was averaged from the indicated number of cells. (**c**) The relative gain in $\Delta C_M$ with OA over vehicle infusion was significantly higher in SN25 LN$^{GGS}$ expressing cells compared to WT controls. (**B**) Properties of the main amperometric spike, displayed as cumulative frequency distribution (*left panel*) and as cell weighted averages for the indicated parameters (*right panel*, a–d). (**C**) Properties of the prespike foot signal (PS) displayed as cumulative frequency distribution (*left panel*) and as cell weighted averages (*right panel*) for the indicated parameters (a–e). Values are given as mean of median determined from the parameter's frequency distribution for each cell. Data are represented as mean ± SEM and were collected for the following conditions (# of events/ # of cells): WT protein (black, 1969/20), WT + OA (dark green, 2640/21), GFP-SN25 LN$^{GGS}$ (cyan, 968/26) and GFP-SN25 LN$^{GGS}$+ OA (pastel pink, 2296/25). Statistical testing was done by ANOVA and Tukey's post-hoc test.

DOI: https://doi.org/10.7554/eLife.41720.032

The following source data is available for figure 8:

**Source data 1.** Extended statistical data as Microsoft Excel spreadsheet.
DOI: https://doi.org/10.7554/eLife.41720.033

*Snap25*[-/-] cells using electrophysiological techniques that allowed us to study secretion kinetics on the single cell level.

Corroborating the idea of a mechanistically relevant linker, we found that mutation of linker motifs severely impaired secretion. In fact, complete substitution of the linker domain by a flexible peptide produced a fully dysfunctional SNAP-25 mutant (*Figure 1B*), while partial substitutions still dramatically reduced primed pool size and fusion rate (*Figure 3A,B*). Albeit large linker substitutions could potentially result in a globally misfolded protein causing unspecific secretory defects, several lines of evidence argue against such scenario: [1] All linker mutants formed SNARE complexes in vitro, showing that both SNARE motifs remain functional (*Figure 2A*, *Figure 3—figure supplement 1C*). [2] The phenotypes of the large substitutions SN25 LN[GGS] and SN25 LC[GGS] qualitatively reflect the deficits of corresponding 'small' mutants SN25 LN[4CS] and SN25 LC[4G]/LC[9G] (*Figure 3*; *Figure 3—figure supplement 2*), implying incremental changes in specific mechanistic processes. [3] Intracellular application of membrane-modifying agents rescued key phenotypical features in SN25 LN[GGS]–expressing cells (*Figures 7* and *8*). If the mutations would just unspecifically interfere with complex nucleation and initial priming, the resulting phenotype should not be highly sensitive to membrane-active compounds that primarily reduce the energy-demand of membrane merger in final fusion steps.

As especially noticeable for the N- and C-terminal mutants SN25 LN[GGS] and SN25 LC[GGS] (*Figure 3A–B*), large linker substitutions caused very similar phenotypes despite targeting non-overlapping regions. While several findings indeed suggest a differential mechanistic function of the N- and C-terminal linker (see discussion below), the apparent convergence of phenotypes likely reflects a structural interdependence between certain motifs due to the relative short length of the linker peptide. Given that the linker motifs are part of the same stretched-out peptide chain that lines the core complex, a potential linker dislocation at one end may well result in an altered configuration at the other end, in this way possibly affecting positioning and function of distant linker sections. The view that the linker is indeed not completely randomly oriented along the SNARE core but at least in sections engages in local interactions with the SNARE complex is supported by recent work of *White et al. (2018)*, showing a diffuse density for the SNAP-25 linker at the N-terminal and C-terminal ends of the EM structure of the fully formed complex.

Taken together, the native linker provides critical functional support for the fusion process, in particular in late mechanistic steps, wherein its motifs stabilize primed vesicles and facilitate fusion triggering. While previous experiments with SNAP-23/25 chimeric proteins only hinted at a regulatory role of a singular C-terminal linker motif (*Nagy et al., 2008*), our new mutational analysis ultimately identifies the SNAP-25 linker as a vital component of the fusion machinery.

## The SNAP-25 Linker is required for productive SNARE interactions

How does the linker domain mechanistically promote secretion? Our biochemical experiments demonstrated that normal SNAP-25:stx-1A interactions critically depend on motifs within the linker (*Figure 2B*), uncovering a putative key aspect of linker function. Since earlier interaction studies could only detect binding between SN1 and stx-1A (*Chapman et al., 1994*; *Hayashi et al., 1994*), the mechanism, by which SN2 enters t-SNARE acceptor complexes, has remained unclear. Here, we provide new insight into initial t-SNARE interplay by showing that a continuous linker-SN2 fragment (but not isolated SN2 motifs) binds to immobilized stx-1A in the presence of free SN1, even increasing the relative retention of SN1 (*Figure 2Be,h*). In reverse, attachment of the linker to SN1 does not allow for binding of SN2 to stx-1A (*Figure 2Bd*), which suggests that only the continuous linker-SN2 fragment acts as a functional unit. We therefore conclude that motifs within the linker peptide are essential for induction and/or stabilization of interactions between SN2 and independently forming SN1:stx-1A complexes. Our experiments with partial substitution mutants moreover revealed that specifically the C-terminal linker segment promotes stx-1A interactions (*Figure 3—figure supplement 1D*). An attractive interpretation would be that local interactions of the C-terminal linker region with the initial segment of SN2 increase its helicity, enabling the association of SN2 with the bi-helical SN1:stx-1A H3 complex (*Figure 9*). As stx-1A:SNAP-25 complexes seem to dynamically alternate between conformational states with different engagement of SN1 and SN2 (*Weninger et al., 2008*), the observed linker-mediated recruitment of SN2 should critically determine conformational dynamics and lifetime of binary complexes. In accord with this conclusion, experiments with SNAP-25-based intramolecular FRET sensors indicated that the two SNARE motifs of SNAP-25 would only

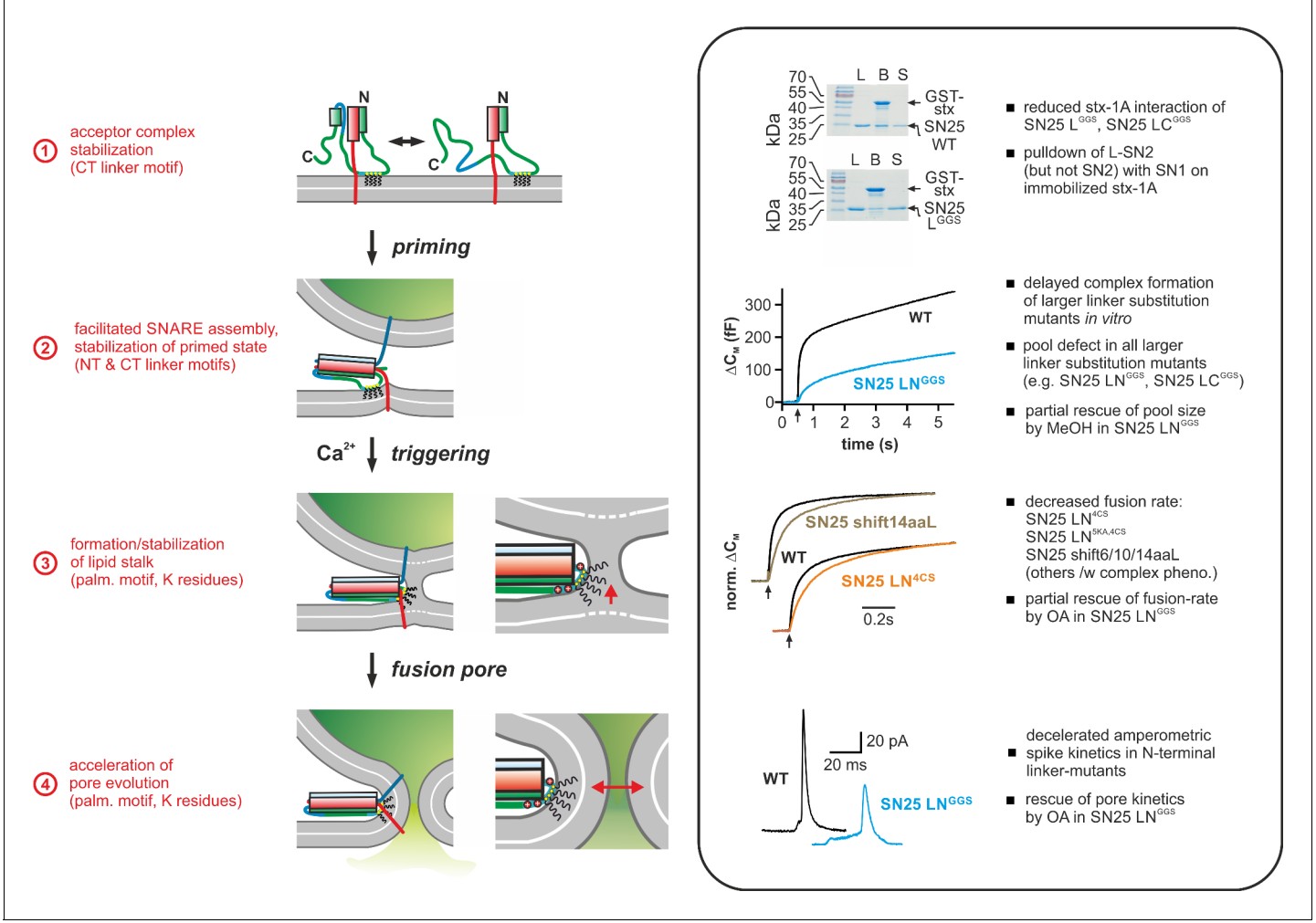

**Figure 9.** Mechanistic model of the role of the SNAP-25 linker domain in exocytosis. Overview of the putative mechanistic functions of the SNAP-25 linker (*cartoon, left side*) with reference to the phenotypic features of linker mutants analyzed in this work (*right side, box*). The cartoon depicts key assembly states of a SNARE complex and potential correlated interactions of the SNAP-25 linker with core complex and/or plasma membrane. Syb-2 is shown in blue, stx-1A in red, and SNAP-25 in green. The acylated N-terminal (NT) linker segment is indicated by the cyan region; the blue section represents the backfolding C-terminal (CT) loop of SNAP-25. Helical segments are shown as boxes. Yellow circles indicate cysteine residues that were modified by acylation (black wavy lines). Positively charged lysine residues are indicated by red circles in the magnified view (for simplicity, only the SNARE motifs of syb-2 and stx-1A are shown in the close-up). Box contains original data (taken from *Figure 2B*, *Figure 3A*, *Figure 5B/C*, and *Figure 6*) in simplified form.

DOI: https://doi.org/10.7554/eLife.41720.034

tightly associate with stx-1a on the plasma membrane, if physically connected by the linker (*Wang et al., 2008*).

On a related note, it is interesting that the retention of SN25 L$^{GGS}$ on immobilized stx-1A closely matched the relative molar binding of isolated SN1 fragments, rendering it highly probable that SN25 L$^{GGS}$ only interacts with stx-1A via its SN1 motif. This provides a ready explanation for the dramatic secretion phenotype of SN25 L$^{GGS}$–expressing KO cells, as binary acceptor complexes, which putatively serve as platforms for syb-2 interactions and vesicle priming (*Fasshauer and Margittai, 2004*; *Pobbati et al., 2006*), would be severely diminished. Our finding that SN25 L$^{GGS}$ decelerates ternary complex assembly in biochemical experiments with free SNARE proteins (*Figure 2A*) would also be consistent with a scenario, in which complex nucleation is compromised by altered stx-1A interactions of SN25 L$^{GGS}$. That said, we cannot exclude that formation of competing unproductive intermediates, e.g. complexes containing only the SN1 motifs together with stx-1A H3 (cf. *Misura et al., 2001*), may contribute to the slowdown of assembly kinetic in vitro and the severe

secretion phenotype of SN25 $L^{GGS}$ in cells. Moreover, linker interactions with the core complex might also facilitate assembly after the initial nucleation stage, as even the N-terminal substitution mutant SN25 $LN^{GGS}$ exhibited a mild slow-down of ternary complex assembly in spite of normal binding to stx-1A (*Figure 3—figure supplement 1C,D*). In line with this idea, MD simulations have indeed suggested potential stabilizing interactions of the linker with the SNARE core complex (*Shi et al., 2017*).

## N-terminal linker:membrane interplay accelerates Ca$^{2+}$-triggered fusion

Attenuated SNARE interactions of linker mutants might indirectly decelerate secretion kinetics by causing a shortage of *trans*-SNARE complexes and an altered stoichiometry of SNARE supercomplexes at the vesicular contact site (*Acuna et al., 2014*; *Mohrmann and Sørensen, 2012*; *Mostafavi et al., 2017*). That said, our structure-function analysis also provided strong evidence for a direct mechanistic involvement of the N-terminal linker region in fusion triggering. While *Nagy et al. (2008)* found that a palmitoylation-deficient SNAP-25 mutant induced a slow-down of secretion, vaguely hinting at a mechanistic requirement for linker acylation, it has remained unclear to what extent the phenotype was affected by the apparent mislocalization of the mutant protein. Here, we have demonstrated that retargeting of the acylation-deficient mutant SN25 $LN^{4CS}$ to the plasma membrane via a secondary palmitoylation site (fused to the GFP-tag) did not mitigate the primary release defects of the mutant (*Figure 4B*), thus excluding the possibility that a reduced membrane expression of SN25 $LN^{4CS}$ per se is responsible for defective fusion triggering. Rather, our data emphasize the requirement of a correctly positioned membrane anchor within the linker of SNAP-25. In this respect, it is highly interesting that the palmitoylation motif directly adjoins the C-terminal end of the SN1 motif throughout all known SNAP-25 orthologs of the animal kingdom, which places the membrane anchor invariantly near the prospective fusion site at the C-terminal end of the assembled SNARE complex, where it might interact with membrane matrices during fusion.

An attractive explanation for the triggering phenotype of SN25 $LN^{4CS}$ would be that the acyl anchors of the linker relay mechanical force from the core complex onto the plasma membrane, thereby increasing local membrane straining. To test this idea, we produced a set of linker mutants, in which the mechanical coupling between SN1 and the palmitoylation motif was loosened by interjacent insertion of flexible spacer peptides. While similar-sized extensions of the JMD of stx or syb-2 dramatically diminished vesicle fusion (*Deák et al., 2006*; *Kesavan et al., 2007*; *Wang et al., 2001*), spacer insertion at the N-terminal end of the linker in SNAP-25 only resulted in a moderate triggering defect with reduced release rates and an increased exocytotic delay (*Figure 5B*), which is clearly distinct from the phenotype observed for the non-acylated SN25 $LN^{4CS}$ mutant. Raising even more doubt about a potential role of the acylated linker in force transfer, we found that insertion of the largest spacer peptide into the palmitoylation-deficient mutant SNAP-25 $LN^{4CS}$ further reduced release rates (*Figure 5C*). Since the linker acyl anchors serve as the primary mechanical coupling point to the plasma membrane, their loss should have vastly precluded any effect of upstream spacer insertion. The considerably exacerbated phenotype of SN25 LN shift14aaLN$^{4CS}$ however indicates that spacer insertion at least in part disrupts an acylation-independent fusion-facilitating function of the N-terminal linker region. In line with this idea, we observed that substitution of several lysine residues, which are thought to mediate very weak electrostatic membrane contacts of the N-terminal linker segment (*Weber et al., 2017*), further decelerated release kinetics in SN25 $LN^{5KA,4CS}$ (*Figure 5—figure supplement 2*). Considering the very moderate defects of spacer insertion mutants and the substantial contribution of acylation-independent effects, it stands to reason that force transfer is not the primary function of linker-mediated membrane interactions. The profound secretory deficits of the non-palmitoylated SN25 $LN^{4CS}$ variant and the observed acylation-independent function of the N-terminal linker segment would rather point to a scenario, in which the palmitoylation motif and the surrounding linker region establish an interface between the SNARE complex and the plasma membrane that directly facilitates triggering by local linker:lipid contacts.

## Mechanistic importance of linker:lipid interactions in membrane fusion

Membrane merger is putatively initiated by spontaneous lipid tail protrusions (lipid splay) near apposition points with reduced hydration or within curved membrane segments (*Smirnova et al., 2010*; *Stevens et al., 2003*; *Tahir et al., 2016*), nucleating an expanding stalk that transforms into a lipidic

fusion pore (*Kawamoto and Shinoda, 2014*; *Risselada et al., 2014*; *Risselada et al., 2011*). Intriguingly, simulations have recently indicated an increased propensity for lipid splay near membrane-embedded viral fusion peptides (*Apelláñiz et al., 2014*; *Kasson et al., 2010*; *Larsson and Kasson, 2013*), suggesting that local protein:lipid interactions are crucial for membrane merger during viral fusion. Moreover, transmembrane domains (TMDs) of SNARE proteins have also been suspected to promote spontaneous stalk formation by disruption of lipid packing in adjoining membrane sections (*Dhara et al., 2016*; *Risselada et al., 2011*). Given these ideas about the mechanistic role of protein:lipid interfaces in fusion, it might be speculated that N-terminal linker:membrane contacts could also be involved in initiation of membrane merger. To test this notion we tried to 'rescue' the secretory phenotype of SN25 LN^GGS by membrane-active compounds that putatively facilitate lipid stalk formation. While it is expected that such treatments also facilitate release in controls, a functional compensation should manifest in a disproportionally strong recovery of release in KO cells expressing the linker mutant. Indeed, we found that the secretion deficits of SN25 LN^GGS can be partially 'rescued' by intracellular application of MeOH or OA, which are both known to alter bilayer properties:

1. Infusion of small amounts of MeOH into KO cells expressing SN25 LN^GGS produced an overly strong increase of total release in comparison to WT controls (*Figure 7B*), indicating a partial recovery of secretion. Short-chain alcohols were previously shown to exert a facilitating effect on the hemifusion/fusion of liposomes (*Chanturiya et al., 1999*; *Paxman et al., 2017*). Although the underlying mechanism is not yet fully understood, it has been argued that alcohol partitioning at the water:lipid interface alters the properties of lipid bilayers, resulting in a functionally relevant reduction in hydration repulsive pressure and interfacial tension (*Ly and Longo, 2004*; *Paxman et al., 2017*; *Stetter and Hugel, 2013*). In our experiments with SN25 LN^GGS, MeOH application conspicuously increased the size of both primed pool components and sustained release, but was unable to elevate release rates of RRP/SRP vesicles. Given the current mechanistic ideas about alcohol-induced effects, a reduced hydration repulsion may ease the convergence of membranes and alleviate the priming defect induced by SN25 LN^GGS. This interpretation implies that the linker:lipid interactions contribute to the dehydration and shielding of phospholipid head groups and thereby energetically promote priming. Indeed, several positively charged residues framing the palmitoylation motif might contribute to this mechanistic function.

2. Infusion of the curvature-inducing lipid OA also overly boosted total secretion in SN25 LN^GGS-expressing cells, indicating a partial compensation of release deficits. Curvature-perturbing lipids are well known to either inhibit or facilitate liposome fusion, viral membrane fusion, and $Ca^{2+}$-dependent exocytosis (reviewed in *Chernomordik and Kozlov, 2003*; *Zhang and Jackson, 2010*). The bidirectional actions of curvature-inducing agents have been largely attributed to the de-/stabilization of curvature-sensitive lipidic fusion intermediates, in accord with recent simulations that report a strong curvature dependence of the energy landscape of membrane fusion (*Kawamoto et al., 2015*). Given the observed recovery of secretion in SN25 LN^GGS-expressing cells after *cis*-application of OA, it seems natural to assume that the linker:lipid interface could also be required for the establishment and/or maintenance of curved membrane segments in critical fusion intermediates. Interestingly, recent studies on S-acylated proteins have suggested that protein palmitoylation does not only allow for curvature-dependent sorting (*Hatzakis et al., 2009*; *Larsen et al., 2015*) but may itself induce membrane curvature (*Chlanda et al., 2017*; *Weber et al., 2017*). In the same line, MD simulations indicated that the palmitoyl lipid anchors of SNAP-25 may sequester cone-shaped cholesterol and stabilize local undulations of the plasma membrane near t-SNARE assemblies (*Sharma and Lindau, 2017*). Hence, it might be speculated that the acylated N-terminal linker supports highly-bent bilayers segments in plasma membrane protrusions or lipid stalks (*Figure 9*).

While the compensatory actions of membrane-active compounds hint at a specific role of linker:lipid interactions in fusion initiation, it should be noted that the available evidence for this scenario is still circumstantial, and other mechanistic interpretations may also apply. In particular, a supporting function of the N-terminal linker in the putative 'zippering' of the JMDs of stx-1A and syb-2 (*Stein et al., 2009*) may represent an alternative explanation, as the linker would be closely involved in inducing membrane merger without directly affecting bilayer properties.

## A postfusional role of linker:lipid interactions in controlling pore dilation

After initial membrane merger, *cis*-SNARE complexes putatively line the neck of the fusion pore, allowing for a tight contact between the palmitoylation motif and the curved flanks of the lipidic pore (*Figure 9*). Here, we have demonstrated that substitution of the N-terminal linker region (but not the C-terminal segment) delayed the discharge kinetics of catecholamines from single vesicles, suggesting that linker:membrane interactions actively control the speed of fusion pore dilation (*Figure 6*). While mutation of the cysteine residues in the palmitoylation motif selectively increased the length of the PS (*Nagy et al., 2008*), we found that perturbations of the whole palmitoylation motif and surrounding sequences (either by direct substitution in SN25 LN[GGS] or by a positional shift in SN25 shift14aa) additionally decelerated the kinetics of the main spike signal. Interestingly, we could demonstrate that this slow-down of the main spike signal is primarily caused by the loss of several lysine residues surrounding the central cysteine group, as substitution of these amino acids in SN25 LN[5KA,4CS] fully replicated the phenotype of the full substitution mutant SN25 LN[GGS] (*Figure 6*). Hence, our data indicate that different linker:lipid interactions are required for subsequent phases of pore expansion: transition from a semi-stable nascent pore to a dilating pore depends on the membrane-embedded acyl anchors of the linker, whereas phospholipid interplay with positively charged residues in the linker is needed for the late phase of pore expansion. Possibly, the changing relevance of different linker:lipid interactions is caused by the shifting geometry of the evolving pore.

It is conspicuous that the phenotypic alterations with respect to the overall secretory response and the transmitter discharge kinetics from single vesicles appear uncorrelated in N- and C-terminal linker mutants. Despite sharing the same kinetic phenotype in $Ca^{2+}$-uncaging experiments, SN25 LN[GGS] and SN25 LC[GGS] strictly differ in fusion pore behavior. As mentioned before, we attribute the convergence of secretion phenotypes in linker mutants to a structural interdependence of different linker motifs during SNARE complex formation. Fusion pore evolution, however, constitutes the final step of the fusion process, when SNARE assembly is almost complete, and indirect effects of the C-terminal linker should be largely diminished. Accordingly, only the direct dependence of pore dilation on N-terminal linker interactions with the plasma membrane should remain relevant.

The decelerating effect of SN25 LN[GGS] on spike kinetics was completely reversed by *cis*-application of cone-shaped OA (*Figure 8B–C*), identifying membrane curvature as a critical factor for linker-mediated regulation of fusion pore expansion. Indeed, theoretical considerations of membrane bending energies explained different pore dilation states by unequal extension mobilities and suggested a strict dependence of pore evolution on membrane tension as well as spontaneous curvature (*Chizmadzhev et al., 1995*). Using a related theoretical approach to deduce the energy state of a metastable fusion pore, *Zhang and Jackson (2010)* also identified membrane curvature as an important determinant of nascent pore stability. Interestingly, based on the assumed relationship between energy landscape and pore life time, a changed PS duration in cells expressing syb-2 TMD mutants was attributed to TMD-induced local curvature changes (*Chang and Jackson, 2015*). Given these ideas about curvature and pore behavior, it seems plausible that the N-terminal acyl anchors of the linker could regulate pore dilation by locally inducing/stabilizing negative membrane curvature. Moreover, we speculate that interactions of the lysine residues in the N-terminal linker region could lower the energetic toll of phospholipid head group crowding in pore segments with highly negative curvature, which would deliver a ready explanation for the additional deceleration of main spike kinetic in cells expressing SN25 LN[5KA,4CS].

## Conclusion

In summary, our data establish a radically changed view of the mechanistic role of the SNAP-25 linker in fast $Ca^{2+}$-triggered exocytosis, acknowledging facilitating functions of linker motifs in multiple mechanistic steps *en route* to fusion (*Figure 9*). In particular, our structure-function analysis revealed two intertwined mechanistic roles of the linker domain: First, the linker serves as an organizer for early t-SNARE interactions and SNARE complex assembly, thus promoting priming and fusion by protein:protein interplay. Second, the acylated N-terminal linker segment engages in membrane contacts that are critical for the maintenance of primed fusion intermediates and fusion triggering. Strikingly, we deliver first clues that linker:lipid contacts facilitate fusion triggering by promoting a favorable membrane configuration for membrane merger. In addition, we demonstrate

that linker:lipid contacts are crucial for fast fusion pore evolution, implying that interactions between the linker and highly-curved membrane segments of the pore may facilitate its expansion. In effect, the multi-tiered facilitating functions of the SNAP-25 linker enhance the performance of the exocytotic machinery, enabling it to meet the speed and fidelity demands of synaptic transmission and neuroendocrine secretion.

# Materials and methods

## Key resources table

| Reagent type (species) or resource | Designation | Source or reference | Identifiers | Additional information |
|---|---|---|---|---|
| Strain, strain background (Mus Musculus) | C57BL/6 | | | |
| Genetic reagent (Mus Musculus) | *Snap25* null allele | *Washbourne et al., 2001*, Genetic ablation of the t-SNARE SNAP-25 distinguishes mechanisms of neuroexocytosis. Nat Neurosci. 5 (1):19–26. PMID:11753414 | $Snap25^{tm1Mcw}$, MGI:2180178 | |
| cDNA (Rattus Norvegicus) | *Snap25a* WT | *Sorensen et al., 2002*, The SNARE protein SNAP-25 is linked to fast calcium triggering of exocytosis.Proc Natl Acad Sci U S A 99(9):6449. PMID: 11830673 | | amino acid sequence identical to murine *Snap25a* |
| Antibody | mouse anti-SNAP-25, SySy 71.1 | Synaptic Systems | Cat# 111 011 | WB 1:1000 |
| Antibody | HRP-conjugated goat-anti mouse | Bio-Rad Laboratories | Cat# 170–5047 | WB 1:5000 |
| Antibody | rabbit anti-SNAP-25, AB1762 | Sigma Adrich | AB1762 | WB 1:1000 |
| Antibody | HRP-conjugated goat-anti rabbit | Bio-Rad Laboratories | Cat# 170–5046 | WB 1:5000 |
| Transfected construct (Mus Musculus) | pSFV1 GFP-SNAP-25 WT | *Sorensen et al., 2002*, The SNARE protein SNAP-25 is linked to fast calcium triggering of exocytosis.Proc Natl Acad Sci U S A 99(9):6449. PMID: 11830673 | | contains rat *Snap25a* WT cDNA; also abbreviated as pSFV1 GFP-SN25 WT |
| Transfected construct (Mus Musculus) | pSFV1 GFP-SN25 $L^{GGS}$ | this paper | | derived from pSFV1 GFP-SNAP-25 WT, substitution of the linker by varied G/S sequence |
| Transfected construct (Mus Musculus) | pSFV1 GFP-SN25 $LN^{GGS}$ | this paper | | derived from pSFV1 GFP-SNAP-25 WT, substitution of the NT linker (aa 83–118) by varied G/S sequence |
| Transfected construct (Mus Musculus) | pSFV1 GFP-SN25 $LC^{GGS}$ | this paper | | derived from pSFV1 GFP-SNAP-25 WT, substitution of the CT linker (aa 119–141) by varied G/S sequence |
| Transfected construct (Mus Musculus) | pSFV1 GFP-SN25 $LC^{4G}$ | this paper | | derived from pSFV1 GFP-SNAP-25 WT, CT substitution $F^{133}G$, $I^{134}G$, $V^{137}G$, $A^{141}G$ |

*Continued on next page*

*Continued*

| Reagent type (species) or resource | Designation | Source or reference | Identifiers | Additional information |
|---|---|---|---|---|
| Transfected construct (Mus Musculus) | pSFV1 GFP-SN25 LC$^{9G}$ | this paper | | derived from pSFV1 GFP-SNAP-25 WT, substitution of F$^{133}$-A$^{141}$ by glycine residues |
| Transfected construct (Mus Musculus) | pSFV1 GFP-SN25 LN$^{4CS}$ | this paper | | derived from pSFV1 GFP-SNAP-25 WT, substitution of 4x cysteine by serine |
| Transfected construct (Mus Musculus) | pSFV1 P-GFP-SN25 WT | this paper | | derived from pSFV1 GFP-SNAP-25 WT, insertion of the minimal SNAP-25 palmitoylation sequence (aa 83–120) NT of GFP |
| Transfected construct (Mus Musculus) | pSFV1 P-GFP-SN25 LN$^{4CS}$ | this paper | | derived from pSFV1 GFP-SN25 LN$^{4CS}$, insertion of the minimal SNAP-25 palmitoylation sequence (aa 83–120) NT of GFP |
| Transfected construct (Mus Musculus) | pSFV1 GFP-SN25 shift6aaL | this paper | | derived from pSFV1 GFP-SNAP-25 WT, insertion of a six aa G/S spacer after G$^{82}$ |
| Transfected construct (Mus Musculus) | pSFV1 GFP-SN25 shift10aaL | this paper | | derived from pSFV1 GFP-SNAP-25 WT, insertion of a 10 aa G/S spacer after G$^{82}$ |
| Transfected construct (Mus Musculus) | pSFV1 GFP-SN25 shift14aaL | this paper | | derived from pSFV1 GFP-SNAP-25 WT, insertion of a 14 aa G/S spacer after G$^{82}$ |
| Transfected construct (Mus Musculus) | pSFV1 GFP-SN25 Lext7aa | this paper | | derived from pSFV1 GFP-SNAP-25 WT, NT insertion of 7 aa analogous to linker of SNAP-23 |
| Transfected construct (Mus Musculus) | pSFV1 GFP-SN25 Lext14aa | this paper | | derived from pSFV1 GFP-SN25 Lext7aa, insertion of a 7aa G/S spacer after A$^{119}$ |
| Transfected construct (Mus Musculus) | pSFV1 GFP-SN25 shift14aaLN$^{4CS}$ | this paper | | derived from pSFV1 GFP-SN25 shift14aaL, substitution of 4x cysteine by serine |
| Transfected construct (Mus Musculus) | pSFV1 GFP-SN25 LN$^{5KA,4CS}$ | this paper | | derived from pSFV1 GFP-SN25 LN$^{4CS}$, substitution K$^{83}$A, K$^{94}$A, K$^{96}$A, K$^{102}$A, K$^{103}$A |
| Transfected construct (Mus Musculus) | pSFV1 GFP-SN1-L IRES SN2 | this paper | | bicistronic expression vector (emcv IRES), newly cloned from pSFV1 and *Snap25* cDNA fragments (SN1-L: aa 1–141; SN2: aa 138–206) |
| Transfected construct (Mus Musculus) | pSFV1 mCherry-SN2 IRES SN1-L | this paper | | bicistronic expression vector (emcv IRES), newly cloned from pSFV1 and *Snap25* cDNA (SN1-L: aa 1–141; SN2: aa 138–206) |
| Transfected construct (Mus Musculus) | pSFV1 mCherry-SN2 IRES SN25Δ26 | this paper | | bicistronic expression vector (polio virus IRES), newly cloned from pSFV1 and *Snap25* cDNA fragments (SN25Δ26: aa 1–180; SN2: aa 138–206) |
| Recombinant DNA reagent | pET28 SN25WT | this paper | | 6xHis-tagged SNAP-25a WT for protein expression |
| Recombinant DNA reagent | pET28 SN25 L$^{GGS}$ | this paper | | 6xHis-tagged SN25 L$^{GGS}$ for protein expression |

*Continued on next page*

*Continued*

| Reagent type (species) or resource | Designation | Source or reference | Identifiers | Additional information |
|---|---|---|---|---|
| Recombinant DNA reagent | pET28 SN25 LN$^{GGS}$ | this paper | | 6xHis-tagged SN25 LN$^{GGS}$ for protein expression |
| Recombinant DNA reagent | pET28 SN25 LC$^{GGS}$ | this paper | | 6xHis-tagged SN25 LC$^{GGS}$ for protein expression |
| Recombinant DNA reagent | pET28 SN1 | this paper | | 6xHis-tagged SN1 (aa 1–82) fragment for protein expression |
| Recombinant DNA reagent | pET28 SN2 | this paper | | 6xHis-tagged SN2 (aa 142–206) fragment for protein expression |
| Recombinant DNA reagent | pET28 SN1-L | this paper | | 6xHis-tagged SN1-L (aa 1–142) fragment for protein expression |
| Recombinant DNA reagent | pET28 L-SN2 | this paper | | 6xHis-tagged L-SN2 (aa 83–206) fragment for protein expression |
| Software, algorithm | IgorPro | WaveMetrics Software | | |
| Software, algorithm | AutesP | NPI electronics | | |
| Software, algorithm | ImageJ | NIH software | | |

## Mutagenesis and viral expression

Substitution or insertion mutations in SNAP-25a were generated by overlap extension polymerase chain reaction (PCR) using appropriate primers containing the desired non-homologous sequences. For expression of the mutant proteins in chromaffin cells, the produced PCR fragments were cloned into a Semliki Forest shuttle vector (pSFV1, Invitrogen) using NsiI or NsiI/BssHII sites for insertion. All SNAP-25 variants were N-terminally fused to GFP using a spacer of 25 amino acids in order to allow for identification of infected cells and quantification of expression levels. To generate bicistronic expression vectors an internal ribosomal entry site (IRES) and a second ORF were inserted into pSFV1 using a BssHII restriction site. In order to yield similar or disparate expression levels of both products, either an encephalomyocarditis virus IRES or a polio virus IRES were used to drive the second ORF. To recognize infected cells, the SN2 fragment was N-terminally tagged with mCherry using a spacer of 18 amino acids. SF viruses were produced using standard protocols. SNAP-25 fragments and linker mutants were also subcloned in pET28a vectors using NheI/BamHII restriction sites for the production of recombinant proteins.

## Cell culture and electrophysiological recordings

Adrenal glands from E18/19 mouse embryos of both genders were used for isolation of chromaffin cells following standard protocols. All mice were handled in compliance with the federal German animal welfare act and local regulations of the University of Saarland. Cultured chromaffin cells were infected with SF viruses on the second and third *day in vitro*, and electrophysiological recordings were performed 5–6 hr post infection. Cultures derived from a given *Snap25$^{-/-}$* animal were equally committed to all experimental conditions. Recorded cells for each tested mutant were derived from ≥2 KO animals. Secretion was characterized by capacitance measurements and simultaneous amperometric recordings as described earlier (*Mohrmann et al., 2013*). Similar numbers of recordings were performed for all conditions during individual sessions. The order, in which different mutants and controls were measured within a session, was varied to avoid a bias in the age of the cultures. Catecholamine release was stimulated either by direct infusion with intracellular solution containing 19 µM free $Ca^{2+}$ via the patch pipette or by photolytic $Ca^{2+}$-uncaging after infusion of nitrophenyl-EGTA (Synaptic Systems). In uncaging experiments, photo-artifacts were removed from amperometric traces by subtraction of a reference measurement with only the carbon fiber electrode in the field of view. Ratiometric $Ca^{2+}$-imaging with fura-4F and furaptra (Invitrogen) was performed as reported before (*Mohrmann et al., 2013*). $[Ca^{2+}]_i$ was raised to 800–900 nM by transient illumination of the cell before a UV flash was applied. The intracellular solution for $Ca^{2+}$-uncaging

experiments was composed of (in mM): 100 Cs-glutamate, 32 Cs-HEPES, 8 NaCl, 4 CaCl$_2$, 2 Mg-ATP, 0.3 GTP, 5 Nitrophenyl-EGTA, 0.4 fura-4F, 0.4 furaptra, and 1 Vitamin C, pH 7.2 (osmolarity adjusted to 290 mOsm). For Ca$^{2+}$-infusion experiments a modified intracellular solution was used (in mM): 110 Cs-glutamate, 8 NaCl, 20 diethylene triamine penta-acetic acid, 5 CaCl$_2$, 2 MgATP, 0.3 Na$_2$GTP, and 40 Hepes-CsOH, pH 7.3 (osmolarity adjusted to 290 mOsm). The extracellular solution contained (in mM): 145 NaCl, 2.8 KCl, 2 CaCl$_2$, 1 MgCl$_2$, 10 HEPES, pH 7.2 (osmolarity adjusted to 300 mOsm). For intracellular infusion of oleic acid (OA), a 2.5 mM stock solution was prepared in DMSO, from which a final concentration of 5 µM OA (1:500 dilution) was prepared in 19 µM free Ca$^{2+}$ containing intracellular solution. For vehicle controls, the same volume of DMSO (without OA) was added to the intracellular solution (0.2% final concentration of DMSO). Membrane capacitance and amperometric spikes were measured simultaneously for 120 s.

Electrophysiological data were analyzed using custom-written macros in IGOR Pro (v. 6.22A, Wavemetrics). Primed vesicle pools and the corresponding release rates were estimated by approximating capacitance traces by three exponential functions. Generally, the two fast exponential functions describe vesicle release from the *readily-releasable pool* ($\tau_{RRP}$<50ms) and the *slowly-releasable pool* (50 ms<$\tau_{SRP}$<500ms), while sustained release is represented by the third slow component. In case that the algorithm delivered a negative amplitude for one component or the fast time constants were not separated by a factor of at least two, the capacitance change was fitted by a biexponential function. Noteworthy, the different kinetic components cannot always be clearly distinguished, when pool sizes and release rates are simultaneously affected.

Single amperometric spikes were recorded using home-made carbon fiber (ø 5 µm, Amoco) electrodes. Digitized amperometric currents were filtered at 3 kHz prior to analysis. Only amperometric spikes with a peak amplitude >4 pA and within the charge range from 10 to 5000 fC were considered for frequency analysis. Amperometric events with a peak amplitude >7 pA were selected for the analysis of kinetic properties using custom macros (Autesp). To gain reliable data on current fluctuations in the pre-spike signal, we restricted our analysis to prespike currents with a total duration longer than 2 ms. For the analysis of prespike signal flickers, the current derivative was filtered at 1.2 kHz. Only deflections exceeding a threshold level of ±6 pA/ms (corresponding to 4*SD) were counted, and fluctuation frequency was calculated as the number of detected suprathreshold fluctuations divided by prespike signal length. Due to the large number of experimental conditions, different sets of mutants were tested in separate experiments. Data were pooled after confirmation that control experiments delivered statistically consistent results.

## Localization and expression analyses

The subcellular localization of GFP-tagged SNAP-25 mutants was investigated 5–6 hr post infection using a laser scanning microscope (LSM 710, Carl Zeiss Microimaging GmbH). Confocal images near the equatorial plane of live cells were acquired at 25°C using a 100x/1.4 oil-objective (Zeiss Plan-Apochromat) with 488 nm excitation, pinhole size of 1 airy unit, and fixed gain settings. Protein distribution near the membrane was quantified by radial line scans covering a distance of ~1.5 µm on both sides of the plasma membrane (ImageJ). Resulting fluorescence profiles were aligned using the maximum of the first derivative, that is the position of maximal fluorescence gain, as anchor point (custom macro, IGOR, Wavemetrics). The position of the plasma membrane was determined based on the clearly detectable fluorescence maximum of GFP-SN25 WT, and membrane expression of mutant proteins was quantified at the corresponding position.

For quantification of total protein expression, epifluorescence pictures of live cells were acquired at 25°C using a 100x/1.3 NA oil-objective (Zeiss) on an Axiovert 25-microscope equipped with a CCD camera (AxioCam MRm). GFP fluorescence throughout the cell body was quantified using ImageJ. Protein levels of SNAP-25 fragments expressed in *Snap25*$^{-/-}$ chromaffin cells by bicistronic SFV were quantified in Western Blot experiments. Primary antibodies specifically recognizing an N-terminal epitope (71.1, Synaptic Systems) or C-terminal epitope (AB1762, Millipore) were employed to quantify fragments using a standard chemoluminescence detection system (Pierce ECL substrate kit). To calculate expression ratios, the concentration of each fragment was determined by a three-point calibration curve, in which defined amounts of bacterially-purified full length protein were used as reference.

## Biochemical analyses

His-tagged SNAP-25 mutants were bacterially expressed (*E. coli* BL21DE3) and purified using a nickel-nitrilotriacetic acid column (Qiagen), as previously described (*Mohrmann et al., 2013*). To investigate ternary SNARE complex formation SNAP-25 variant, stx-1A (aa 1–262), and GST-syb-2 (aa 1–116) were first dialyzed into a solution containing (in mM): 100 NaCl, 20 Tris, 1 DTT, 1 EDTA, 0.5% Triton X-100, pH 7.4. Equal amounts (~3 µM) of SNAP-25/mutant variant, stx-1A, and GST-Syb-2 were mixed in a test volume of 80 µl and incubated for different time intervals at 25°C under slight agitation. SNARE assembly was stopped by addition of SDS-containing sample buffer, and (unboiled) samples were analyzed by SDS-PAGE and Coomassie blue staining of protein bands. Gels were scanned on a flat-bed scanner, and SNARE complex formation was quantified by densitometry using ImageJ.

For stx-1A binding assays with SNAP-25 fragments and linker mutants, GST-stx-1a (aa 1–262, 25 µM) was first immobilized on glutathione-sepharose beads (2 hr, 4°C). After extensive washing, beads were incubated for 20 hr at 4°C with test proteins (5 µM) in a binding buffer containing 100 mM NaCl, 20 mM Tris, 1 mM DTT, 1 mM EDTA, 0.5% Triton X-100 (pH 7.4). Subsequently, beads were washed four times, resuspended in fresh binding buffer, and protein retention was analyzed by SDS-PAGE and Coomassie blue staining. The amount of bound protein was quantified by densitometry using ImageJ. Assuming that Coomassie staining is directly proportional to protein size under our conditions, we calculated the molar ratio $n_{bound}/n_{stx1A}$ accounting for the different molecular mass of test proteins.

## Statistics

Electrophysiological measurements of secretion in individual cells were considered as independent biological replicates throughout this study. In all electrophysiological experiments, we aimed for a minimal sample size of >10 independent recordings. The actual number of replicates (*n*) is provided in the corresponding figure panels. Balanced numbers of recordings for all experimental conditions were acquired in each recording session. In biochemical assays, data from individual samples of mixed purified proteins were handled as independent biological replicates for statistical comparisons. Data were excluded from analysis, if common technical standards in individual experiments were violated (e.g. unstable electrophysiological recordings with high leak current). We did not use a statistical method to identify and eliminate 'outliers'. Statistical tests were performed in Sigmaplot 12 (Systat Software). Following standard procedures, we generally used unpaired Student's t-test for comparisons between two experimental conditions and ANOVA for experiments featuring multiple conditions. For specific comparisons between groups after ANOVA, we employed Tukey's post-hoc test, if not stated otherwise. Extended statistical data are provided for each figure as Source Data. Significance levels: '*' for $p < 0.05$, '**' for $p < 0.01$, and '***' for $p < 0.001$.

## Acknowledgements

The authors are grateful to N Vogel, D Alansary, B Niemeyer, J Rettig, and N Brose for valuable discussions and support. We would like to thank M Wirth, B Bimperling, and V Schmitt for excellent technical assistance. The work was supported by grants from the Deutsche Forschungsgemeinschaft to RM (MO2312/1-1, SFB 1027) and DB (SFB 1027). The authors declare no conflict of interest.

## Additional information

### Funding

| Funder | Grant reference number | Author |
| --- | --- | --- |
| Deutsche Forschungsge-meinschaft | MO2312/1-1 | Ralf Mohrmann |
| Deutsche Forschungsge-meinschaft | SFB 1027 | Dieter Bruns<br>Ralf Mohrmann |

The funders had no role in study design, data collection and interpretation, or the decision to submit the work for publication.

## Author contributions
Ahmed Shaaban, Madhurima Dhara, Formal analysis, Validation, Investigation, Visualization, Writing—review and editing; Walentina Frisch, Ali Harb, Ali H Shaib, Formal analysis, Investigation; Ute Becherer, Conceptualization, Resources, Methodology, Writing—review and editing; Dieter Bruns, Conceptualization, Resources, Funding acquisition, Methodology, Writing—review and editing; Ralf Mohrmann, Conceptualization, Software, Formal analysis, Supervision, Funding acquisition, Investigation, Writing—original draft, Project administration, Writing—review and editing

## Author ORCIDs
Madhurima Dhara  http://orcid.org/0000-0001-7745-472X
Dieter Bruns  https://orcid.org/0000-0002-2497-1878
Ralf Mohrmann  https://orcid.org/0000-0001-9279-5071

## Ethics
Animal experimentation: Breeding Snap25 KO mice was done in compliance with the German animal welfare act and all local regulations of the Saarland University and the Otto-von-Guericke-University Magdeburg. Animals were only used for post portem tissue extraction (no experimentation on alive animals).

## Decision letter and Author response
Decision letter https://doi.org/10.7554/eLife.41720.038
Author response https://doi.org/10.7554/eLife.41720.039

# Additional files

## Supplementary files
• Transparent reporting form
DOI: https://doi.org/10.7554/eLife.41720.035

## Data availability
All data generated or analysed during this study are included in the manuscript and supporting files.

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
