## [Decision Letter]

Thank you for submitting your article "The SNAP-25 linker supports fusion intermediates by local lipid interactions" for consideration by *eLife*. Your article has been reviewed by Randy Schekman as the Senior Editor, a Reviewing Editor, and three reviewers. The reviewers have opted to remain anonymous.

The reviewers have discussed the reviews with one another and the Reviewing Editor has drafted this decision to help you prepare a revised submission.

Summary:

In this manuscript, the authors use mutagenesis, secretion measurements from mouse adrenal chromaffin cells, and in vitro SNARE assembly and pulldown methods to re-investigate the role of the linker region, which connects the two SNARE-motifs in SNAP-25 (denoted SN1 and SN2). They identify two different functions for the N- and C-terminal end of the linker. The C-terminus of the linker is found to stabilize the syntaxin-SNAP-25 acceptor complex for VAMP/synaptobrevin, whereas the palmitoylated N-terminus of the linker supports fast triggering, priming and fusion pore expansion. Interestingly, the effect of replacing the N-terminus of the linker (including the palmitoylated cysteines) with a flexible GGS-motif is overcome by infusion of oleic acid, which promotes curvature due to its cone-shape.

The authors investigate SNARE complex formation in vitro using split SNAP25 constructs and show convincingly that syntaxin binds the constructs SN1 plus linker-SN2 and that the presence of the linker on SN2 is essential. These results point to a role of the linker in efficient SNARE complex assembly. Replacing the linker within unsplit SNAP25 by GGS repeats with the same total length also supports SNARE complex formation, although with only 50% efficiency compared to wt SNAP25. However, this construct does not support fusion in response to caged calcium stimulation as assessed by capacitance measurements and amperometry. Mutation of the 4 cysteins to serine reduces fusion amplitude and slows down the kinetics. Adding an extra palmitoylation site linked to the N terminus does provide plasma membrane localization but the 4CS mutation still reduces fusion amplitude and speed, suggesting that the location of the palmitoylation site matters. To address this question the authors performed experiments introducing GGS linkers of variable length between SN1 and the Palmitoylation sites. These mutations have rather small impact on fusion. The amplitudes are unchanged and only small changes in kinetics are observed.

There are several concerns and suggestions (detailed below) that should be addressed in a revised manuscript. Moreover, the presentation requires revision.

Essential revisions:

The author's statement "The aggravated phenotype of SN25 shift14aaLN4CS strongly argues against a significant contribution of the linker to force transfer but rather suggests that delayed fusion is caused by a misalignment of membrane and protein interactions of the N-terminal linker section." Is unclear and questionable. First, the impaired fusion kinetics of the LN-4CS, the shift6aaL, shift10aaL and shift14aaL render the opposite conclusion likely (that the force transfer IS impaired) – how else do the authors explain the phenotypes? Moreover, the rationale for this experiment is apparently that the 4CS-mutation has already eliminated the possibility for force transduction to the membrane, and therefore the exacerbation based on the 14 aa insertion must be due to something else. However, the authors themselves show in Figure 5—figure supplement 2 that removal of lysine residues on the background of the 4CS-mutation lead to similar changes (longer time constants of RRP release), and the authors conclude "the lysine residues contribute to the linker:lipid interactions that facilitates Ca^2+^-dependent fusion triggering". Since the lysines are still present in the shift14aaLN-4CS mutation this finding undermines the conclusion that the linker is not conferring force between the SNAREs and the lipid membrane. Another point is that the insertion of 14aa close to the nascent fusion pore could have dominant-negative effects, which might affect fusion. It is also not clear why "misalignment of membrane and protein interactions" is an alternative to impaired force transfer. Would such a misalignment not affect force transfer? Given that the authors did not do mutations to change the force released by the SNAREs, the conclusion about the transfer of force should be tempered.

What is the reason to use both methanol and oleic acid to rescue the LN-GGS? Methanol seems to have almost rescued the uncaging-induced secretion amplitude, but not the time constants of release, whereas oleic acid rescues the shape of amperometric spikes. Does this represent two different phenotypes, or only one? I.e. did methanol not rescue the amperometric spike shape, and was this the reason that oleic acid was then used? (And vice versa?)

Apparently, the authors conclude that the two ends of the SNAP-25 linker subserves two very different functions: the C-terminal part is involved in supporting t-SNARE interactions, whereas the N-terminal linker with its acyl side-chains 'engage in local lipid interactions'. However, why are the phenotypes of the LN-GGS and the LC-GGS then basically indistinguishable in uncaging experiments (Figure 2)? Does this not undermine the conclusion? Can the phenotype of the LC-GGS in uncaging experiments be rescued by methanol and oleic acid?

Mutation of the 4 cysteins to serine reduces fusion amplitude and slows down the kinetics but to a lesser degree than the full replacement with SGG repeats or SGG replacement in the N terminal residues 82-118 of the linker. Very similar results are obtained for corresponding mutations in the C terminal parts of the linker (residues 118-142), leaving the palmitoylation sites intact. It would be interesting to know if these constructs are actually palmitoylated in the cells or if there are convincing reasons to assume that they are. Maybe this is shown in Figure S4 Ai but needs to be stated specifically in the main text.

In Figure 4Bii it is impossible to tell which trace is which. There are (presumably) 3 black control traces with slightly different kinetics and 2 or 3 "shift" traces which are slightly slower but also very similar. As presented, it is unclear what can actually be concluded form this data. Also, the rationale of using a highly flexible linker rather than a rigid linker is not obvious because it raises the question how much the Palmitoylation sites are spatially displaced from their wt location.

Would it be possible to generate an LC4C construct where a palmitoylation site is introduced at the N terminal and of the linker next to SN2?

Surprisingly, Figure 4C shows that the effect of 14aaL is much more pronounced in the LN4CS mutant, which lacks the palmitoyl anchor. Doesn't this result suggest that the lipid anchors stabilize a reasonable SNARE complex structure, making it rather resistant to the introduced spacers and that in the absence of this stabilization the extra spacers have more detrimental impact on the SNARE complex?

In the Discussion section this leads to contradicting statements: "insertion at the N terminal end…… decrease release kinetics irrespective of membrane anchorage" and "…the phenotypes……reflect functional deficits originating from misalignment of the Palmitoylation site with other components of the fusion machinery".

The analysis of quantal events in Figure 5 is very interesting. However, it needs to be explained how PS amplitude and initial PS amplitude were defined. An example of spike analysis illustrating the different parameters would help. What is the significance of the observed changes in PS amplitude values? Do they suggest a change in fusion pore structure?

It is interesting that LC-GGS is indistinguishable from wild type, while there is very strong reduction in amplitude and kinetics of secretion in the flash experiment. This should be discussed more specifically.

For the results of Figure 6 and Figure 7, effects of methanol and oleic acid, the discussion is a bit misleading. The authors point to a rescue effect of the LN-GGS mutation, but it seems that these compounds similarly affect the results in normal wild type cells. It is not convincingly explained why these specifically relate to a rescue effect of the LN-GGS mutation.

The authors suggest that the linker domain of SNAP-25 prevents dimerization of the SNARE complexes "by masking the sticky surfaces". However, there is no direct proof for that except an apparent size shift in the SDS-PAGE. Inspecting the Figure 1C and Figure 3—figure supplement 1C carefully, one can see that also with WT SNAP-25 construct, a band with a similar size-shift becomes visible at 24hours. Additionally, even though it's apparent from the reduction of the monomeric SNAREs that the high molecular mass represents the SNARE complex, the identity of this high molecular mass band is not clear, because it's just Coomassie staining. Especially in the case of separated SN1 and SN2 fragments (Figure 1C), as there is no obvious reduction in the SN fragments and also as SN2 cannot even bind to Stx1 (Figure 1D), it cannot be concluded that high-size band represents dimerized SNARE complexes. This part should be toned down.

Using the fragments SN1+L-SN2 for SNARE complex assembly would also be informative, as in this case the interaction of SN2 with Stx1 is partially rescued and that in turn could rescue Syb2 binding as well. Last, it is not very clear what the star in Figure 1D stands for, and also why there are two fragments when SNAP-25 LGGS is used. The lower band seems at the size of SN1-L. Is the SNAP-25 LGGS construct somewhat unstable and degraded?

The expression experiments using the SNAP-25 linker mutants are not very clear (Figure 1—figure supplement 3A,B). The total expression of these mutants might be comparable to WT, but it is not informative for the plasma membrane localization (except Figure 3—figure supplement 1A) However, the interpretation of the functional output using the mislocalized proteins should be done more cautiously. Also, a test for proper localization regarding the 5KA,4CS mutant as in Figure 3—figure supplement 1A would be useful. Removal of 5K in the absence of palmitoylation could further affect the membrane targeting and localization of SNAP-25, which would explain the further reduction of the release (Figure 5A). Finally, the release properties of the cells expressing only the 5KA mutant should be examined because a mislocalization of this mutant is less likely due to proper palmitoylation of the cysteines. This would be useful for understanding whether the effects of 5KA and 4CS mutations are additive.

The authors claim in subsection “N- and C-terminal linker regions mechanistically support Ca^2+^-dependent release”, that SNAP-25 LC4G mutant shows phenotypical traits of SNAP-25 LCGGS. However, SNAP-25 LC4G shows only an intermediate phenotype. This claim should be therefore toned down. Comparing the functional output of mislocalized proteins creates ambiguity. The authors mention in subsection “A postfusional role of linker:lipid interactions in controlling pore dilation” that "this slow-down of the main spike signal is primarily caused by the loss of several lysine residues surrounding the central cysteine group, as substitution of these amino acids in SN25 LN5KA,4CS fully replicated the phenotype of the full substitution mutant SN25 LNGGS." However, the resemblance between the phenotypes of SNAP25-LNGGS and LN5KA,4CS might well be merely due to lack of membrane attachment and thus mislocalization. A membrane targeting strategy similar to what is done in Figure 3 would be more informative regarding the 4CS mutants combined with linker insertion (14aa; Figure 4C) and 5KA mutants (Figure 5—figure supplement 2).

Addition of another membrane targeting motif (Figure 3) could potentially change the conformation of SNAP-25. Besides, comparing the release parameters in Figure 3 and in the other figures, it seems that this P-GFP-SNAP25 WT has a better release efficacy. This is a very minor point, but could be mentioned in discussion.

The interpretation of MeOH and oleic acid experiments should be toned down. It is clear that MeOH and OA facilitate release, but this could be due to a mechanism independent from the function of the linker region. In other words, MeOH or OA already increases and accelerates the NT release from cells expressing WT SNAP-25, and as the cells expressing the linker mutant already has deficits in NT release, it is not surprising that they show a higher sensitivity to MeOH and OA application. Even though the data is largely convincing, the possibility of independent action of MeOH or OA from linker-lipid interaction cannot be ruled out.

In further support of a structural significance of the SNAP-25 linker (rather than being a "flexible linker"), a recent higher resolution EM structure of the NSF/aSNAP/SNARE complex revealed diffuse density for the SNAP-25 linker at both the N-terminal and C-terminal ends, suggesting that these parts of the linker are associated with the SNARE complex, albeit probably sampling multiple conformations. Thus, the linker is not entirely flexible, but it is not uniquely structured either (see Figure 7 in White et al., 2018). Perhaps this recent finding could be added to the Discussion section.

---

## [Author Response]

Essential revisions:1) The author's statement "The aggravated phenotype of SN25 shift14aaLN4CS strongly argues against a significant contribution of the linker to force transfer but rather suggests that delayed fusion is caused by a misalignment of membrane and protein interactions of the N-terminal linker section." Is unclear and questionable. First, the impaired fusion kinetics of the LN-4CS, the shift6aaL, shift10aaL and shift14aaL render the opposite conclusion likely (that the force transfer IS impaired) – how else do the authors explain the phenotypes? Moreover, the rationale for this experiment is apparently that the 4CS-mutation has already eliminated the possibility for force transduction to the membrane, and therefore the exacerbation based on the 14 aa insertion must be due to something else. However, the authors themselves show in Figure 5—figure supplement 2 that removal of lysine residues on the background of the 4CS-mutation lead to similar changes (longer time constants of RRP release), and the authors conclude "the lysine residues contribute to the linker:lipid interactions that facilitates Ca^2+^-dependent fusion triggering". Since the lysines are still present in the shift14aaLN-4CS mutation this finding undermines the conclusion that the linker is not conferring force between the SNAREs and the lipid membrane. Another point is that the insertion of 14aa close to the nascent fusion pore could have dominant-negative effects, which might affect fusion. It is also not clear why "misalignment of membrane and protein interactions" is an alternative to impaired force transfer. Would such a misalignment not affect force transfer? Given that the authors did not do mutations to change the force released by the SNAREs, the conclusion about the transfer of force should be tempered.

We thank the reviewers for pointing out that our conclusions with regard to the spacer insertion mutants were imprecisely formulated. Due to length concerns, we tried to keep explanations very concise, which may have hurt accessibility and clarity. While we acknowledge the reviewers’ doubts about the correct interpretation of the mutant phenotypes, we still think that our findings render it unlikely that the acylated linker plays a prominent role in force transduction. Earlier work has demonstrated that extension of the juxtamembrane domain of synaptobrevin or syntaxin critically interferes with liposome fusion, vacuole fusion in yeast, and Ca^2+^-triggered secretion due to impaired force transfer (Deak et al., 2006; Guzman et al., 2010; Kesavan et al., 2007; McNew et al., 1999; Wang et al., 2001). In particular, it has been established that insertion of flexible spacers >=12 aa (high Gly-content) abolishes membrane fusion in most model systems (Deak et al., 2006; Guzman et al., 2010; Kesavan et al., 2007). Assuming that an effective cooperation between SNAP-25 and its cognate SNARE partners requires a similarly tight coupling of all membrane anchoring structures to the core complex, it is expected that the insertion of 14 amino acids before the acylation site is sufficient to severely compromise linker-mediated force transfer and to induce secretion defects similar to the non-acylated SN25 LN^4CS^ mutant. However, cells expressing SN25 shift14aaL exhibited unexpectedly weak secretory deficits (Figure 5B), thus questioning the idea that the acylated linker has a substantial contribution to force transfer.

The notion of linker-mediated force transduction is further challenged by our finding that insertion of the 14 aa-long spacer into SN25 LN^4CS^ exacerbated its phenotype (Figure 5C). The rationale behind this experiment was that elimination of membrane anchorage should abolish force transfer and thus should preclude any additional effects of spacer insertions on force transduction. The more pronounced kinetic slowdown of secretion in the combined mutant however implies that spacer insertion interferes with mechanisms that do not depend on linker acylation. As insinuated by the reviewer, it might be argued that elimination of acylation in SN25 LN^4CS^ does not abolish all lipid interactions, and that the residual membrane contacts still play a role in force transfer. While membrane binding energies for unstructured peptides are generally considered to be negligibly small (Almeida, 2014; White and Wimley, 1999), recent work by Weber et al., (2017) indeed suggested that an excess of positive charged lysine residues around the cysteine cluster in the N-terminal could mediate transient membrane contacts. However, the membrane free binding energy of short peptides containing 5 lysine residues (which is comparable to the 5 lysine residues in the N-terminal linker segment) has been estimated to be just around -6 kJ/mol (Kim et al., 1991). In comparison, a crude assessment of the Gibbs free energy gain for the membrane insertion of the four palmitate moieties of SNAP-25 yields an maximal energy gain of -150kJ/mol according to Tanford’s formula (Pool and Thompson, 1998; Tanford, 1980). MD simulations have even estimated binding free energies of ~ 250 kJ/mol for regular TMDs (Chetwynd et al., 2010; Lindau et al., 2012). From the perspective of binding energies, it is thus implausible how the very weak electrostatic membrane association of SNAP-25 could significantly contribute to force transfer, as mechanical tension would likely disrupt these interactions before a membrane deformation occurs. In line with the labile nature of the electrostatic linker:membrane contacts, previous studies also reported very low binding of non-palmitoylated SNAP-25 to membrane sheets (Nagy et al., 2008; Weber et al., 2017).

While the weak lipid interactions of the flanking lysine residues cannot relay pulling force on the plasma membrane, their elimination still caused a slight but noticeable slowdown of secretion kinetics in SN25 LN^5KA,4CS^ (Figure 5—figure supplement 2) – similar to the effects seen in spacer insertion mutants. We think that this finding hints at a scenario, in which the acylation site and surrounding residues are part of a membrane-interacting structure that actively induces fusogenicity but does not primarily serve as an anchorage-point for SNARE force transfer to the plasma membrane. Thus, it seems straight forward to assume that the functional defects caused by spacer insertion result from a displacement of this membrane interacting linker structure, which was ambiguously referred to as a “misalignment” of the N-terminal linker segment in the original version of the manuscript. Given that treatment of cells with the membrane-perturbing agents oleic acid and methanol disproportionally facilitated release in cells expressing the N-terminal substitution mutant SN25 LN^GGS^ (Figure 7, Figure 8), we speculate that local membrane interactions of the N-terminal linker structure could indeed directly alter membrane properties to similarly increase fusogenicity. Despite the compensatory action of oleic acid on fusion pore evolution in the mutant (Figure 8), we cannot exclude that the presence of the spacer could also sterically affect the fusion pore in its direct vicinity, as proposed by the reviewer. Nevertheless, such effect would not change the general interpretation of our results with regard to force transfer, as the flexible spacer should still loosen the mechanical coupling to the acylated linker. Hence, one would have expected to find a more pronounced phenotype for SN25 shift14aaL in such scenario, if linker-mediated force transfer were indeed highly relevant.

To accommodate the reviewers’ concerns, we have largely rewritten the corresponding passages in Results section and Discussion section. The relative weakness of the electrostatic membrane interactions of the lysine residues in the linker is now better discussed (subsection “The acylated N-terminal linker segment promotes triggering”), resulting in a clearer line of argument. Following the reviewers’ suggestion, we have also tempered our conclusion about force transfer and now modestly state in the Discussion section:

“Considering the very moderate defects of spacer insertion mutants and the substantial contribution of acylation-independent effects, it stands to reason that force transfer is not the primary function of linker-mediated membrane interactions.”

What is the reason to use both methanol and oleic acid to rescue the LN-GGS? Methanol seems to have almost rescued the uncaging-induced secretion amplitude, but not the time constants of release, whereas oleic acid rescues the shape of amperometric spikes. Does this represent two different phenotypes, or only one? I.e. did methanol not rescue the amperometric spike shape, and was this the reason that oleic acid was then used? (And vice versa?)

We used methanol (MeOH) and oleic acid (OA) in our rescue experiments, because both were reported to facilitate membrane fusion by altering lipid bilayer properties. Their effects on membranes are not congruent, as infusion of cells with MeOH putatively modifies the surface hydration and lipid packing of the exposed leaflet, while OA inclusion primarily induces negative membrane curvature. Earlier preparatory experiments have indicated to us that infusion of MeOH only exerts a strong effect on total secretion but leaves fusion pore evolution largely unaffected. In proof, we provide a dataset illustrating the persistence of spike properties in wildtype (WT) cells infused with 2% MeOH (Author response image 1). In contrast to MeOH treatment, intracellular application of OA was found to generally facilitate fusion rate and to simultaneously accelerate spike kinetics. As we expected that MeOH treatment would not alter spike properties in SN25 LN^GGS^-expressing cells, we did not perform high-resolution amperometric recordings.

**Author response image 1. respfig1:** Effect of intracellular Methanol (MeOH, 2%) application on the properties of amperometric spikes elicited by infusion of 19 µM Ca2+. (**a**) Representative amperometric spikes recorded in WT chromaffin cells in the absence (black) or presence (red) of MeOH. (**b**) Superimposition of averaged spike waveforms from two successively recorded cells, one untreated (113 spikes), the other infused with MeOH (138 spikes). (**c**) Detailed kinetic analysis of the prespike foot signals: Shown are mean foot amplitude, charge, and duration (WT: 2308 events, 20 cells; WT+MeOH: 3730 events, 25 cells). (**d**) Quantitative analysis of main spike parameters: Depicted are mean amplitude, charge, rise time (50-90%), and half-width. No significant differences were observed. Error bars represent SEM.

The results of the rescue experiments with MeOH and OA suggest that linker:membrane contacts are likely important for initial fusogenicity/fusion pore opening and the subsequent postfusional dilatation of the pore. While methanol and OA treatment can both ameliorate total secretion, only the OA-induced curvature seems to be important for the fast expansion of the fusion pore. From this perspective, one can distinguish two phenotypes of SN25 LN^GGS^ in consecutive stages, as suggested by the reviewers.

Apparently, the authors conclude that the two ends of the SNAP-25 linker subserves two very different functions: the C-terminal part is involved in supporting t-SNARE interactions, whereas the N-terminal linker with its acyl side-chains 'engage in local lipid interactions'. However, why are the phenotypes of the LN-GGS and the LC-GGS then basically indistinguishable in uncaging experiments (Figure 2)? Does this not undermine the conclusion? Can the phenotype of the LC-GGS in uncaging experiments be rescued by methanol and oleic acid?

Several of our findings indeed suggest a differential mechanistic function of N- and C-terminal linker motifs in secretion. However, it is also true that we observed phenotypic similarities between different N- and C-terminal linker mutants. While larger linker substitutions invariably induced a “complex” phenotype with priming and fusion triggering deficits of different intensity, only very restricted manipulations within the very N-terminal region (especially in case of the spacer insertion mutants SN25 shift6/10/14aaL) resulted in a mechanistically distinct fusion defect. We think that the occurrence of complex phenotypes most likely reflects a structural interdependence between different motifs due to the relative short length of the linker peptide. New structural data derived from high-resolution cryo-EM experiments have in fact indicated that the linker more closely lines the core complex near its ends (White et al., 2018), implying the existence of interaction sites in both terminal regions. Thus, the substitutions in SN25 LN^GGS^ or SN25 LC^GGS^ should at least interfere with one of these interaction regions. Given that the linker motifs are part of the same stretched-out peptide chain, a potential linker dislocation at one end of the linker may well result in an altered configuration at the other end, in this way possibly affecting the function of almost all linker motifs. To acknowledge this idea, we have added a statement concerning the structural interdependence of motifs to the Results section:

“As especially noticeable for the N- and C-terminal mutants SN25 LN^GGS^ and SN25 LC^GGS^ (Figure 3A-B), large linker substitutions caused very similar phenotypes despite targeting non-overlapping regions. While several findings indeed suggest a differential mechanistic function of the N- and C-terminal linker (see discussion below), the apparent convergence of phenotypes likely reflects a structural interdependence between certain motifs due to the relative short length of the linker peptide. Given that the linker motifs are part of the same stretched-out peptide chain that lines the core complex, a potential linker dislocation at one end of the linker may well result in an altered configuration at the other end, in this way possibly affecting positioning and function of distant linker sections.”

It is an interesting question whether the uncaging phenotype of SN25 LC^GGS^ can be rescued with methanol or OA in a similar way as seen for SN25 LN^GGS^. If our above interpretation is correct that large substitutions can affect N- as well as C-terminal linker motifs, a similar rescue of total secretion for both mutants should be expected. However, due to time restrictions, an experimental confirmation of this aspect was unfortunately beyond the scope of this revision.

Mutation of the 4 cysteins to serine reduces fusion amplitude and slows down the kinetics but to a lesser degree than the full replacement with SGG repeats or SGG replacement in the N terminal residues 82-118 of the linker. Very similar results are obtained for corresponding mutations in the C terminal parts of the linker (residues 118-142), leaving the palmitoylation sites intact. It would be interesting to know if these constructs are actually palmitoylated in the cells or if there are convincing reasons to assume that they are. Maybe this is shown in Figure S4 Ai but needs to be stated specifically in the main text.

We investigated the subcellular distribution of the C-terminal substitution mutant GFP-SN25 LC^GGS^ by confocal laser microscopy and did not find any significant change in its membrane localization compared to the WT protein (shown in Figure 3 —figure supplement 1A). These data clearly suggest an unaltered palmitoylation level of the mutant protein. In contrast to GFP-SN25 LC^GGS^, membrane targeting of the N-terminal substitution mutant GFP-SN25 LN^GGS^, whose palmitoylation motif was ablated, was diminished and mirrored the distribution pattern of the non-palmitolayted GFP-SN25 LN^4CS^ mutant. To complement the existing dataset, we have now also performed a detailed confocal analysis of the other two C-terminal mutants, SN25 LC^4G^ and SN25 LC^9G^, and reconfirmed their normal localization at the plasma membrane (Figure 1—figure supplement 3Bb). We have rewritten the corresponding passage to better convey the information (subsection “N- and C-terminal linker regions mechanistically support Ca^2+^-dependent release”):

“Note that all C-terminal linker mutants were normally targeted to the plasma membrane and exhibited a similar overall expression level like the WT protein in controls (Figure 1—figure supplement 3A,Bb), excluding the possibility that the observed secretion deficits are due to reduced expression.”

In Figure 4Bii it is impossible to tell which trace is which. There are (presumably) 3 black control traces with slightly different kinetics and 2 or 3 "shift" traces which are slightly slower but also very similar. As presented, it is unclear what can actually be concluded form this data.

We thank the reviewers for giving us feedback on the presentation of the normalized capacitance traces in original Figure 4Biii. The panel was indeed intended to highlight the kinetic slowdown in the secretory response of the three spacer insertion mutants, SN25 shif6aa/10aa/14aaL. For this purpose, we subtracted the linear component and normalized all traces to the corresponding ΔC_M_ value 1s after the uncaging flash. As the depicted data was derived from three separate experimental series with independent control recordings, we included the corresponding wildtype rescue traces (black lines) for each dataset but unfortunately did not mark, which control trace referred to which mutant condition. To accommodate the reviewers’ criticism, we now display the three pairs of traces separately (Figure 5Bc).

Also, the rationale of using a highly flexible linker rather than a rigid linker is not obvious because it raises the question how much the Palmitoylation sites are spatially displaced from their wt location.

We intentionally used flexible spacer peptides with high glycine content to distance the SNARE motif from the acylation site, as the high rotational freedom of glycine residues is expected to result in a random coil configuration with reduced conformational strain, which should aggravate the effect of spacer insertion on force transfer. Indeed, previous studies (Deak et al., 2006; Kesavan et al., 2007; McNew et al., 1999), which investigated the role of mechanical coupling by extension of the juxtamembrane domains of synaptobrevin/syntaxin, have exclusively relied on insertion of highly flexible spacers. While a rigid linker may allow for a relatively controlled displacement of the acyl anchors, it should be less efficient in hampering the mechanical coupling to the acyl anchors. Moreover, there is a greater risk to induce unwanted structural issues with respect to other parts of the fusion machinery, as a helical spacer might interact with the preceding amphiphatic SN1 helix or sterically affect the putative “zippering” of the juxtamembrane domains of synaptobrevin/syntaxin in its direct vicinity (Stein et al., 2009). Thus, we are convinced that the used flexible spacers serve our purpose best.

We have slightly altered the corresponding passage in the Results section to better explain the experimental rationale behind the usage of flexible linkers:

“To further test this idea, we constructed linker mutants, in which the distance between the acylation site and SN1 was increased by insertion of flexible spacer peptides (6, 10, or 14 residues; mutants denoted SN25 shift6aa/10aa/14aaL) in order to mechanically “uncouple” the acyl anchors from the core complex (Figure 5A). We especially relied on G/S-containing spacers, as the high rotational freedom of the included Glycine residues should substantially lessen conformational strain and thereby further hamper force transfer.”

Would it be possible to generate an LC4C construct where a palmitoylation site is introduced at the N terminal and of the linker next to SN2?

In principle, a short sequence containing the cluster of cysteine residues (e.g. aa 85-93) could be easily inserted into the linker near the N-terminal end of the complex directly preceding SN2. However, it is not clear whether acylation of the inserted cysteine residues at this position would occur in an efficient manner. Previous truncation experiments in SNAP-25 have identified the N-terminal linker region including aa 85-120 as the minimal palmitoylation motif, thus indicating a requirement for an extended linker segment in acylation (Gonzalo et al., 1999). While the corresponding region is obviously still present in SN25 LN^4CS^ and could even mediate transient plasma membrane contacts as proposed by Weber et al., (2017), it is a concern that the changed relative position of the cysteine cluster may obstruct an efficient interaction with acyltransferases. A further issue arises from our observation that the linker section preceding SN2 is putatively involved in functionally relevant core complex interactions (Figure 3B, Figure 3—figure supplement 1C-D). An insertion of a secondary acylation site in the C-terminal linker region is therefore expected to cause deficits that need to be distinguished from those functional alterations that may originate from altering the position of the acyl anchors.

Considering the potential pitfalls of the experimental approach, we think that such mutant is not ideal. Moreover, we already present data showing that shifting the acylation site to an N-terminal position relative to the complex (P-GFP-SN25 LN^4CS^; Figure 4) resulted in similar deficits as the non-acylated SN25 LN^4CS^ mutant. Clearly, one would expect that the proposed switch of the acyl anchors to the opposite end of the linker would at best lead to a similar result.

Surprisingly, Figure 4C shows that the effect of 14aaL is much more pronounced in the LN4CS mutant, which lacks the palmitoyl anchor. Doesn't this result suggest that the lipid anchors stabilize a reasonable SNARE complex structure, making it rather resistant to the introduced spacers and that in the absence of this stabilization the extra spacers have more detrimental impact on the SNARE complex?

We consider it difficult to adequately compare the phenotypical deficits caused by spacer insertion in SN25 shift14aaL and SN25 shiftLN^4CS^. Consequently, we are skeptical about the reviewers’ conclusion that “the effect of 14aaL is much more pronounced in the LN4CS mutant”. In general, largely disproportional changes of secretion parameters in SN25 shift14aaLN^4CS^-expressing cells should be indicative of a potential synergistic aggravation of the phenotype, as insinuated by the reviewer. Looking at our kinetic data, we however noticed only a ~1.5x increase in τ_RRP_ after spacer insertion in SN25 shift14aaLN^4CS^ (Figure 5Cb) while the corresponding change in τ_RRP_ was 3.5x for SN25 shift14aaL (in comparison to wildtype controls; Figure 5Bb). Moreover, the relative changes in other parameters were actually quite similar: The time constants for SRP-mediated release were extendedby almost the same factor (1.7x), and the exocytotic delay roughly doubled in both cases. Thus, in our opinion, our data do not support the notion that the spacer-induced relative changes in SN25 shift14aaLN^4CS^ were dramatically more pronounced than in SN25 shift14aaL. Our results would rather fit a scenario, in which the spacer insertion more or less proportionally modulates the stronger phenotype of SN25 LN^4CS^, thus providing little support for a stabilizing effect of linker acylation.

In the Discussion section this leads to contradicting statements: "insertion at the N terminal end…… decrease release kinetics irrespective of membrane anchorage" and "…the phenotypes……reflect functional deficits originating from misalignment of the Palmitoylation site with other components of the fusion machinery".

We are sorry that our conclusions concerning the role of the N-terminal linker region seem inconsistent to the reviewer. As already elaborated in response to comment #1, the aggravated phenotype of SN25 shift14aaLN^4CS^ suggests that the triggering defect after spacer insertion is in substantial part due to an acylation-independent mechanism. Given that the acyl anchors are by far the most important lipid-interacting element for mechanical straining of the membrane, we interpret the modest phenotype of SN25 shift14aaL and the enhanced slowdown of secretion in the SN25 shift14aaLN^4CS^ mutant as evidence against a prominent role of the linker in force transfer. Therefore, it stands to reason that the functional deficits of spacer insertion mutants actually result from a displacement of the N-terminal linker region that disrupts its local protein and/or lipid interactions near the C-terminal end of the core complex. In line with this idea, substitution of several lysine residues – which previously have been associated with electrostatic linker:plasma membrane interactions (Weber et al., 2017) – slightly decelerated secretion in SN25 LN^5KA,4CS^ mutant (Figure 5—figure supplement 2).

While the role of linker acylation for efficient fusion triggering is somewhat “downplayed” by the observation that spacer insertion aggravates the phenotype of non-palmitoylated SNAP-25, it should be noted that the profound kinetic phenotype of the non-palmitoylated SN25 LN^4CS^ mutant still indicates that the acyl anchors serve a major function in fusion initiation. Unfortunately, the coexistence of acyl anchor-dependent and acylation-independent mechanisms was not clearly expressed in our original text, leading to ambiguous statements. To clarify this issue, we have now reworked the corresponding passage in the Discussion section. The central statement referred to by the reviewers now reads:

“Since the linker acyl anchors serve as the primary mechanical coupling point to the plasma membrane, their loss should have vastly precluded any effect of upstream spacer insertion. The considerably exacerbated phenotype of SN25 LN shift14aaLN^4CS^ however indicates that spacer insertion at least in part disrupts an acylation-independent fusion-facilitating function of the N-terminal linker region. In line with this idea, we observed that substitution of several lysine residues, which are thought to mediate very weak electrostatic membrane contacts of the N-terminal linker segment (Weber et al., 2017), further decelerated release kinetics in SN25 LN^5KA,4CS^ (Figure 5—figure supplement 2). Considering the very moderate defects of spacer insertion mutants and the substantial contribution of acylation-independent effects, it stands to reason that force transfer is not the primary function of linker-mediated membrane interactions. The profound secretory deficits of the non-palmitoylated SN25 LN^4CS^ variant and the observed acylation-independent function of the N-terminal linker segment would rather point to a scenario, in which the palmitoylation motif and the surrounding linker region establish an interface between the SNARE complex and the plasma membrane that directly facilitates triggering by local linker:lipid contacts.”

The analysis of quantal events in Figure 5 is very interesting. However, it needs to be explained how PS amplitude and initial PS amplitude were defined. An example of spike analysis illustrating the different parameters would help. What is the significance of the observed changes in PS amplitude values? Do they suggest a change in fusion pore structure?

As suggested by the reviewer, we have incorporated an example for the spike analysis in Figure 6. For the sake of simplicity, we have highlighted PS duration, PS amplitude, spike amplitude, spike half width and rise time (50%-90%) in the example trace. The initial PS amplitude is determined at the initiation of the PS event, where the current amplitude is at least 4 times of the baseline noise. As such, the initial PS amplitude indicates the width of the fusion pore at the moment of opening. As shown in the example trace, the PS amplitude denotes the maximum current flow during the initial fusion pore state. A change in this parameter likely mirrors the maximum size of the narrow pore. Both parameters, initial PS amplitude and the PS amplitude, were reduced in cells expressing the GFP-SN25 LN^GGS^ mutant, indicating that the size of the pore at the first moment of opening as well as the maximum size of the fusion pore are significantly reduced by the linker mutation.

It is interesting that LC-GGS is indistinguishable from wild type, while there is very strong reduction in amplitude and kinetics of secretion in the flash experiment. This should be discussed more specifically.

The reviewers obviously refer to our finding that SN25 LC^GGS^-expressing cells exhibit a dramatically reduced secretory response but do not show detectable changes in the kinetics of amperometric spikes. This is especially remarkable, since the N-terminal linker substitution mutant SN25 LN^GGS^ possesses a virtually identical secretion phenotype in flash experiments, but does show a prominent slowdown of spike kinetics in contrast to SN25 LC^GGS^. We have attributed the delayed transmitter discharge in SN25 LN^GGS^–expressing cells to the loss of critical membrane interactions of the N-terminal linker segment, which is left unchanged in SN25 LC^GGS^. Thus, N- and C-terminal linker segments seem to be differentially required in early and late stages of the fusion mechanism.

As mentioned in response to reviewer comment #3, the similarity of phenotypes between N- and C-terminal linker substitution mutants in flash experiments potentially indicates a structural crosstalk of N- and C-terminal linker motifs in defining the exact configuration of the linker along the core complex. Such structural effects should be largely limited to the complex assembly phase, before the linker is fully stretched out. If SNARE assembly is almost complete, the influence of C-terminal linker mutations on the overall linker configuration should cease, while the membrane-anchored, N-terminal linker region should still be able to exert effects in postfusional stages. This would provide a satisfactory explanation for the observed phenotypical difference of both mutants. Following the reviewers’ recommendation, these ideas have been briefly mentioned in the Discussion section:

“It is conspicuous that the phenotypic alterations with respect to the overall secretory response and the transmitter discharge kinetics from single vesicles appear uncorrelated in N- and C-terminal linker mutants. Despite sharing the same kinetic phenotype in Ca^2+^-uncaging experiments, SN25 LN^GGS^ and SN25 LC^GGS^ strictly differ in fusion pore behavior. As mentioned before, we attribute the convergence of secretion phenotypes in linker mutants to a structural interdependence of different linker motifs during SNARE complex formation. Fusion pore evolution, however, constitutes the final step of the fusion process, when SNARE assembly is almost complete, and indirect effects of the C-terminal linker should be largely diminished. Accordingly, only the direct dependence of pore dilation on N-terminal linker interactions with the plasma membrane should remain relevant.”

For the results of Figures 6 and Figure 7, effects of methanol and oleic acid, the discussion is a bit misleading. The authors point to a rescue effect of the LN-GGS mutation, but it seems that these compounds similarly affect the results in normal wild type cells. It is not convincingly explained why these specifically relate to a rescue effect of the LN-GGS mutation.

We generally define a successful “rescue” as a functional compensation resulting from interventions that target and successfully bypass the compromised mechanistic step. While this definition does not require that treatments are functionally neutral in control experiments, we strictly demand that successful interventions induce a substantially stronger enhancement of release in SN25 LN^GGS^-expressing KO cells than in controls. Considering the fusion process as a simplistic linear sequence of mechanistic reaction steps, non-compensating treatments should yield (at best) similar relative effects for mutant and controls, if they target steps upstream or downstream of the compromised process, leaving the mechanistic “bottleneck” unaffected. In contrast, an over-proportional recovery is expected for interventions that can effectively compensate for the compromised function of mutant SNAP-25 in the same mechanistic context. Indeed, we found that intracellular application of methanol or oleic acid resulted in a significantly higher relative increase in total secretion of SN25 LN^GGS^-expressing cells than in control experiments, thus fulfilling our basic criteria for a successful “rescue”. While the observed “rescue” strongly suggests that the N-terminal linker is important for a fusion intermediate that is sensitive to membrane-perturbing agents, our data does not necessarily imply that the membrane effects of the N-terminal linker motif and the compounds are congruent. Nevertheless, we consider it the most likely scenario that the N-terminal linker increases fusogenicity by locally altering membrane properties (like e.g. lipid packing and/or curvature). To improve the clarity of our argumentation we have changed several passages, in particular the Results section:

“If N-terminal linker:membrane interactions critically modulate lipidic fusion intermediates, application of membrane-active reagents might be able to “rescue” release deficits. The underlying idea is that linker:membrane interplay might mechanistically help to facilitate spontaneous lipid stalk formation and membrane merger, and thus might serve a function that could be mimicked by application of membrane-perturbing compounds. While such treatments generally promote fusion (in mutants as well as in controls), the relative effect should be disproportionally strong in KO cells expressing SN25 LN^GGS^, if the action of the compound can lessen the impact of the mechanistic “bottleneck” imposed by the mutation.”

Additionally, we streamlined the corresponding section in the Discussion section and added the following statement to stress the possibility of alternative explanations:

“While the compensatory actions of membrane-active compounds hint at a specific role of linker:lipid interactions in fusion initiation, it should be noted that the available evidence for this scenario is still circumstantial, and other mechanistic explanations might also apply. In particular, a supporting function of the N-terminal linker in the putative “zippering” of the JMDs of stx-1A and syb-2 (Stein et al., 2009) may represent an alternative explanation, as the linker would be closely involved in inducing membrane merger without directly affecting bilayer properties.”

The authors suggest that the linker domain of SNAP-25 prevents dimerization of the SNARE complexes "by masking the sticky surfaces". However, there is no direct proof for that except an apparent size shift in the SDS-PAGE. Inspecting the Figure 1C and Figure 3—figure supplement 1C carefully, one can see that also with WT SNAP-25 construct, a band with a similar size-shift becomes visible at 24hours. Additionally, even though it's apparent from the reduction of the monomeric SNAREs that the high molecular mass represents the SNARE complex, the identity of this high molecular mass band is not clear, because it's just Coomassie staining. Especially in the case of separated SN1 and SN2 fragments (Figure 1C), as there is no obvious reduction in the SN fragments and also as SN2 cannot even bind to Stx1 (Figure 1D), it cannot be concluded that high-size band represents dimerized SNARE complexes. This part should be toned down.

The dimerization of SNARE complexes formed out of isolated SN1, SN2, Syntaxin, and Synaptobrevin has been investigated in detail by the workgroup of Prof. Sabine Hilfiker. This group used size-exclusion chromatography, MALLS, and SAXS to characterize SNARE complex dimers and found that two fully formed SNARE complexes can C-terminally attach to each other in a salt-dependent way, forming a wing-shaped dimer (Fdez et al., 2008). While our analysis is rudimentary, we are convinced that the observed size-shifted bands for SNARE complexes employing linker-less SNAP-25 fragments indeed represents the described complex dimers. Moreover, Fasshauer et al., (1998) and others have shown that separated, linker-less SN1 and SN2 assemble into a SDS-stable core complexes, even though SN2 cannot bind directly to stx1 (Chapman et al., 1994). As the increased formation of complex dimers in the presence of SNAP-25 linker mutants in vitro is however not central to this work and admittedly distracts from more important results, we have decided to shorten the corresponding passages in the interest of a streamlined manuscript. Only a basic statement concerning the shift in band size was retained (subsection “The SNAP-25 linker is critical for t-SNARE interaction and complex assembly”):

“SNARE complexes formed by SN25 L^GGS^ or separated SN1 and SN2 motifs exhibited a characteristic size shift (~180 kDa; Figure 2Ab-c), which possibly suggests the formation of SNARE complex dimers. Indeed, previous work by Fdez et al., (2008) has shown that two fully formed ternary complexes containing linker-less SN1/SN2 fragments can assemble into dimeric structures with attached C-terminal tips. Thus, the primary structure of the SNAP-25 linker peptide likely affects the biochemical properties of SNAREpins and their propensity to dimerize.”

Using the fragments SN1+L-SN2 for SNARE complex assembly would also be informative, as in this case the interaction of SN2 with Stx1 is partially rescued and that in turn could rescue Syb2 binding as well.

We have followed the reviewers’ recommendation and tested SNARE complex formation for a combination of SN1 and L-SN2 fragments (Figure 2—figure supplement 1). Although we had trouble to correctly match the amount of the small fragments in all experiments, we noticed a significant acceleration of complex assembly in comparison to controls using linker-less SN1 and SN2 fragments. That said, it needs to be taken into account that the “inverse” setup using SN1-L and SN2 also positively affected SNARE complex formation in parallel experiments, indicating that linker:core complex interactions may additionally exert a promoting effect on complex assembly. Our biochemical assays presumably underestimate the effect of t-SNARE interactions on the assembly kinetics, as fragments can easily interact with each other without steric hindrance in solution. A corresponding passage describing our findings has been added to the Results section:

“As the requirement for a continuous linker-SN2 segment in t-SNARE interactions might limit ternary complex nucleation, we also compared SNARE complex assembly in mixtures containing either SN1-L and SN2, SN1 and L-SN2, or linker-less SN1 and SN2 together with stx-1A and GST-syb-2 (Figure 2 – —figure supplement 1A). We observed that the presence of the linker in either fragment combination significantly facilitated SNARE assembly over linker-less fragments, as judged by the time constants of mono-exponential fits of assembly profiles (Figure 2—figure supplement 1B). Intriguingly, the combination of SN1/L-SN2-fragments was slightly more efficient than SN1-L/SN2 in promoting complex formation, indicating that linker interactions during initial t-SNARE assembly might indeed determine overall assembly kinetics. That said, our biochemical assays likely underestimate the effect of t-SNARE interactions, as fragments can easily interact with each other in solution without steric hindrance.”

Last, it is not very clear what the star in Figure 1D stands for, and also why there are two fragments when SNAP-25 LGGS is used. The lower band seems at the size of SN1-L. Is the SNAP-25 LGGS construct somewhat unstable and degraded?

We noticed a contaminating protein (band marked by “*”) in the purified fraction of bacterially expressed His_6_-SN25 L^GGS^ that likely represents a partially degraded expression product. Since only low levels of the contaminating protein were present, and further optimization of the purification procedure failed to completely eliminate the side product, we used the resulting protein fraction despite the presence of the contamination. We do not expect any effect of the contamination on our results, as we did neither see a participation of the protein in complex formation nor in stx-1A-binding. A corresponding statement has been added to the legend of Figure 2.

The expression experiments using the SNAP-25 linker mutants are not very clear (Figure 1—figure supplement 3A,B). The total expression of these mutants might be comparable to WT, but it is not informative for the plasma membrane localization (except Figure 3—figure supplement 1A) However, the interpretation of the functional output using the mislocalized proteins should be done more cautiously. Also, a test for proper localization regarding the 5KA,4CS mutant as in Figure S4A would be useful. Removal of 5K in the absence of palmitoylation could further affect the membrane targeting and localization of SNAP-25, which would explain the further reduction of the release (Figure 5A).

We thank the reviewers for reminding us of the importance of reliable expression data for a valid interpretation of mutant phenotypes. To eliminate remaining uncertainties concerning the membrane expression of SNAP-25 linker mutants, we performed additional confocal analyses to study the membrane localization of our GFP-tagged SNAP-25 variants. These new expression data has been added as Figure 1—figure supplement 2C and Figure 1—figure supplement 3Ba-c. Interestingly, we did not find any additional decrease in the membrane expression of the SN25 LN^5KA,4CS^ mutant that could account for the slight exacerbation of its phenotype in comparison to SN25 LN^4CS^. Accordingly, it must be concluded that local phospholipid interaction of the lysine residues play a mechanistic role in fusion initiation. Moreover, we did not observe any effect of spacer insertions on the membrane localization of the corresponding mutants, confirming that the observed phenotypes are not due to a lowered availability of the mutant protein on the plasma membrane.

Finally, the release properties of the cells expressing only the 5KA mutant should be examined because a mislocalization of this mutant is less likely due to proper palmitoylation of the cysteines. This would be useful for understanding whether the effects of 5KA and 4CS mutations are additive.

We agree with the reviewers that the role of the lysine residues in the N-terminal linker deserves more attention, as we might underestimate their function, when only looking at the combined SN25 LN^5KA,4CS^ mutant. However, Weber et al., (2017)recently showed that these residues mediate transient membrane interactions that are important for acylation of SNAP-25 by membrane-bound DHHC acyl-transferases. This implies that mutation of the lysine residues should always produce effects on the acylation state of SNAP-25. Due to this complication, we consider it not worthwhile at this point to investigate the 5KA-mutation in isolation.

The authors claim in subsection “N- and C-terminal linker regions mechanistically support Ca^2+^-dependent release”, that SNAP-25 LC4G mutant shows phenotypical traits of SNAP-25 LCGGS. However, SNAP-25 LC4G shows only an intermediate phenotype. This claim should be therefore toned down. Comparing the functional output of mislocalized proteins creates ambiguity.

We are sorry for the ambiguous formulation. The corresponding sentence was intended to stress that both phenotypes are related and share certain key features, but it should not express that both mutants exhibit identical release deficits. As correctly stated by the reviewer, SN25 LC^4G^ (and also SN25 LC^9G^) shows an intermediate phenotype in comparison to the larger substitution mutant SN25 LC^GGS^. The passage has been accordingly altered to (subsection “N- and C-terminal linker regions mechanistically support Ca^2+^-dependent release”):

“Hence, this small substitution in SN25 LC^4G^ already induced a complex release phenotype that is reminiscent of the defects seen in SN25 LC^GGS^.”

The authors mention in subsection “A postfusional role of linker:lipid interactions in controlling pore dilation” that "this slow-down of the main spike signal is primarily caused by the loss of several lysine residues surrounding the central cysteine group, as substitution of these amino acids in SN25 LN5KA,4CS fully replicated the phenotype of the full substitution mutant SN25 LNGGS." However, the resemblance between the phenotypes of SNAP25-LNGGS and LN5KA,4CS might well be merely due to lack of membrane attachment and thus mislocalization. A membrane targeting strategy similar to what is done in Figure 3 would be more informative regarding the 4CS mutants combined with linker insertion (14aa; Figure 4C) and 5KA mutants (Figure 5—figure supplement 2).

We agree with the reviewers that the membrane expression of the SNAP-25 linker mutants might critically affect release properties, especially with respect to fusion pore behavior, which has recently been proposed to depend on SNARE stoichiometry (e.g. (Shi et al., 2012; Wu et al., 2017)). To investigate whether different membrane expression levels may account for phenotypical differences between related linker mutants, we have now performed an extended analysis of protein distribution using confocal microscopy. In particular, we deliver better information on the membrane expression of SN25 LN^5KA4CS^ in comparison to SN25 LN^4CS^. As illustrated in Figure 1—figure supplement 3Ba,we did not observe any differences in the distribution profiles of both mutants, which indicates that electrostatic interactions of the N-terminal linker with phospholipids are only of minor importance for the steady-state level of non-palmitoylated SNAP-25 at the membrane. Moreover, our older data on the localization of SN25 LN^GGS^ demonstrated that the larger substitution within the N-terminal linker region did not result in a more pronounced mislocalization in comparison to SN25 LN^4CS^ (Figure 3—figure supplement 1A). With all three N-terminal linker mutants sharing a similar low membrane expression, there is currently no indication that the observed phenotypic difference in spike kinetics (SN25 LN^4CS^ exhibits only prespike signal alterations, while SN25 LN^5KA4CS^ and SN25 LN^GGS^ also show a marked delay in main spike kinetics) are the consequence of differential membrane targeting of the mutant proteins. While very small changes may have escaped our confocal analysis, it would be hard to explain by stoichiometric changes why a dramatic decrease in SNAP-25 abundance would only affect the prespike signal in SN25 LN^4CS^, and a subtle further drop should account for the dramatically delayed main spike kinetic in SN25 LN5^KA,4CS^. Thus, we think the phenotypic differences are indeed the consequence of a differential interference with linker:lipid interactions.

While it surely is an interesting idea to analyze the functional effects of lysine substitution (5KA) in front of the retargeted P-GFP-SN25 LN^4CS^ variant, we think that our dataset on membrane expression already firmly excludes a prominent influence of membrane targeting on the phenotypic features in SN25 LN^5KA,4CS^ (s. discussion above). Similarly, we also do not see how plasma membrane expression could influence the phenotypical features of spacer insertion (shift14L), as normal membrane targeting was seen in SN25 shift14aaL (Figure 1—figure supplement 3Bc), and residual membrane expression was unaffected after spacer insertion in SN25 LN^4CS^ (Figure 1—figure supplement 3Ba). Due to the high work effort and the relatively low gain in information, we decided against testing the “5KA” and “shift14aa” mutations in P-GFP-SN25 LN^4CS^.

Addition of another membrane targeting motif (Figure 3) could potentially change the conformation of SNAP-25. Besides, comparing the release parameters in Figure 3 and in the other figures, it seems that this P-GFP-SNAP25 WT has a better release efficacy. This is a very minor point, but could be mentioned in discussion.

We think that the higher secretory responses are a consequence of the slightly higher expression of P-GFP-SNAP-25 WT at the plasma membrane. As mentioning this point would have disrupted the line of argument in the Discussion section, we have added a corresponding statement to the Results section instead:

“P-GFP-SN25 WT efficiently rescued secretion in KO chromaffin cells and even reached a slightly higher release level than typically seen in other control recordings (Figure 4B). This increase in total secretion might be attributed to a more pronounced membrane targeting compared to the unmodified WT protein.”

The interpretation of MeOH and oleic acid experiments should be toned down. It is clear that MeOH and OA facilitate release, but this could be due to a mechanism independent from the function of the linker region. In other words, MeOH or OA already increases and accelerates the NT release from cells expressing WT SNAP-25, and as the cells expressing the linker mutant already has deficits in NT release, it is not surprising that they show a higher sensitivity to MeOH and OA application. Even though the data is largely convincing, the possibility of independent action of MeOH or OA from linker-lipid interaction cannot be ruled out.

It is true that rescue experiments with membrane–perturbing compounds (OA and MeOH) can only deliver circumstantial evidence with regard to the nature and functional significance of linker-mediated membrane interactions. That said, we do not see compelling reasons to assume that any given mutation generically renders the SNARE machinery highly sensitive to OA or MeOH. We expect that a compound-driven facilitation of secretion should be similar or smaller than the corresponding relative changes in WT cells, if the treatment has no direct effect on the fusion intermediate that depends on the mutated linker region (see our response to comment #12). In this case, the process affected by the linker mutation should still limit secretion, as vesicle fusion putatively constitutes a series of sequential mechanistic steps. Only interventions that actually target the same step or fusion intermediate are expected to lift the blockade and produce a substantial disproportional compensation.

There are of course scenarios, in which the higher responsiveness to the treatments could be due to indirect actions of the linker on the OA/MeOH-sensitive intermediate, and thus might result in a misleading positive “rescue”. This could occur, if the N-terminal linker region tightly supports the force transfer by syb and/or stx (e.g. via facilitates C-terminal core complex assembly and zippering of JMDs) and hence acts as a modulator rather than a direct effector on the lipidic fusion intermediate. In this case, the apparent rescue would provide little information about the role of linker-mediated membrane interactions. As we have not explicitly discussed the possible limitations of the approach in the original manuscript version, we have now added a corresponding statements in the text, stressing that our results hint at but not necessarily prove the proposed role of linker:lipid interaction in membrane merger (subsection “A postfusional role of linker:lipid interactions in controlling pore dilation”):

“While the compensatory actions of membrane-active compounds hint at a specific role of linker:lipid interactions in fusion initiation, it should be noted that the available evidence for this scenario is still circumstantial, and other mechanistic explanations might also apply. In particular, a supporting function of the N-terminal linker in the putative “zippering” of the JMDs of stx-1A and syb-2 (Stein et al., 2009) may represent an alternative explanation, as the linker would be closely involved in inducing membrane merger without directly affecting bilayer properties.”

In further support of a structural significance of the SNAP-25 linker (rather than being a "flexible linker"), a recent higher resolution EM structure of the NSF/aSNAP/SNARE complex revealed diffuse density for the SNAP-25 linker at both the N-terminal and C-terminal ends, suggesting that these parts of the linker are associated with the SNARE complex, albeit probably sampling multiple conformations. Thus, the linker is not entirely flexible, but it is not uniquely structured either (see Figure 7 in White et al., 2018). Perhaps this recent finding could be added to the Discussion section.

We thank the reviewers for bringing this new publication to our attention. These findings have now been mentioned in the Discussion section:

“The view that the linker is indeed not completely randomly oriented along the SNARE core but at least in sections engages in local interactions with the SNARE complex is supported by recent work of White et al., (2018), showing a diffuse density for the SNAP-25 linker at the N-terminal and C-terminal ends of the EM structure of the fully formed complex.”